# Combined modelling of mRNA decay dynamics and single-molecule imaging in the *Drosophila* embryo uncovers a role for P-bodies in 5′ to 3′ degradation

**Lauren Forbes Beadle**[1☉], **Jennifer C. Love**[1☉], **Yuliya Shapovalova**[1¤], **Artem Artemev**[2], **Magnus Rattray**[1]*, **Hilary L. Ashe**[1]*

1 Faculty of Biology, Medicine and Health, University of Manchester, Manchester, United Kingdom,
2 Department of Computing, Imperial College London, London, United Kingdom

☉ These authors contributed equally to this work.
¤ Current address: Institute for Computing and Information Sciences, Radboud University, Nijmegen, the Netherlands
* magnus.rattray@manchester.ac.uk (MR); hilary.ashe@manchester.ac.uk (HLA)

**Data Availability Statement:** RNA-seq data are deposited in ArrayExpress under accession number: E-MTAB-11580. Python implementation

## Abstract

Regulation of mRNA degradation is critical for a diverse array of cellular processes and developmental cell fate decisions. Many methods for determining mRNA half-lives rely on transcriptional inhibition or metabolic labelling. Here, we use a non-invasive method for estimating half-lives for hundreds of mRNAs in the early *Drosophila* embryo. This approach uses the intronic and exonic reads from a total RNA-seq time series and Gaussian process regression to model the dynamics of premature and mature mRNAs. We show how regulation of mRNA stability is used to establish a range of mature mRNA dynamics during embryogenesis, despite shared transcription profiles. Using single-molecule imaging, we provide evidence that, for the mRNAs tested, there is a correlation between short half-life and mRNA association with P-bodies. Moreover, we detect an enrichment of mRNA 3′ ends in P-bodies in the early embryo, consistent with 5′ to 3′ degradation occurring in P-bodies for at least a subset of mRNAs. We discuss our findings in relation to recently published data suggesting that the primary function of P-bodies in other biological contexts is mRNA storage.

## Introduction

Cells establish their identity by changing their gene expression patterns in response to different signals and environments. Critical to this is the ability of a cell to modulate mRNA levels. mRNA abundance depends not only on the transcription rate but also on mRNA stability. In eukaryotic cells, there are 2 major pathways of mRNA degradation: Xrn1 endonuclease-mediated 5′-3′ decay and exosome catalysed 3′-5′ decay [1,2]. Many mRNA degradation factors and mRNAs can become condensed into processing bodies (P-bodies), which are phase separated compartments in the cytoplasm implicated in mRNA storage and decay [3,4]. mRNA stability

of the model is available from: https://github.com/ManchesterBioinference/GP_Transcription_Dynamics. Scripts for image analysis are available from: https://github.com/j-c-love/PLOSBiology_ForbesBeadle_Love_et.al_2022.

**Funding:** This research was funded by a Wellcome Trust Investigator award to H.L.A. and M.R. (204832/Z/16/Z, 204832/B/16/Z) and a Wellcome Trust PhD studentship to J.C.L. (222814/Z/21/Z). The funders had no role in study design, data collection and analysis, decision to publish, or preparation of the manuscript.

**Competing interests:** The authors have declared that no competing interests exist.

**Abbreviations:** AEL, after egg lay; CF, cephalic furrow; CSC, codon stabilisation coefficient; GO, Gene Ontology; GP, Gaussian process; MALA, Metropolis-adjusted Langevin algorithm; MAP, maximum a posteriori; ODE, ordinary differential equation; PCA, principal component analysis; RBF, radial basis function; RPKM, reads per kilobase of transcript per million reads mapped; smFISH, single-molecule fluorescent in situ hybridization; TPM, transcripts per million; TS, transcription site.

is also commonly regulated by sequences in the 3′ UTR, including binding sites for RNA binding proteins or miRNAs [5]. While the 2 major decay pathways are responsible for general turnover of cytoplasmic mRNAs, there are also mRNA surveillance pathways that degrade aberrant mRNAs. These include mRNAs carrying a premature stop codon, lacking a stop codon, or mRNAs with paused ribosomes [6].

Regulation of mRNA degradation is essential for diverse cellular processes including proliferation, differentiation, apoptosis, and immune responses [1,7–10]. Control of mRNA stability is also important for cellular decisions and behaviour during development. For example, regulation of *myc* mRNA stability fine-tunes the proliferation rate of neuroblasts in the *Drosophila* larval brain [11], an *fgf8* mRNA gradient generated by mRNA decay couples differentiation to posterior elongation of the vertebrate embryonic axis [12] and *Hes1* mRNA instability is integral to the Hes1 protein ultradian oscillations that may act as a timer for vertebrate neuronal differentiation [13]. In addition, a conserved feature of early embryogenesis is that there is bulk degradation of maternal mRNAs around the time of zygotic genome activation [14,15]. Consistent with the key roles of mRNA stability in cell biology, mutations in many components of the degradation pathways are associated with human diseases [9,16].

While the half-lives of strictly maternal mRNAs during embryogenesis can be readily measured genome-wide [17], measuring the decay of zygotic mRNAs is more difficult due to ongoing transcription. One approach is to inhibit transcription and then follow the decline in mRNA levels over time [18–20]. Other methods involve metabolic labelling of the RNA, for example, in pulse-chase or approach-to-equilibrium experiments [19–21]. Related approaches use computational models to estimate transcription and degradation rates by sequencing both the total and labelled RNA following the pulse [19]. Single-molecule fluorescent in situ hybridisation (smFISH) imaging-based methods for estimating mRNA half-lives have also been described. However, these methods are not high throughput and require either steady-state transcript levels [22] or a natural shut off of transcription [23].

In this study, we generate a high-resolution total RNA-seq time series across early embryogenesis that we use to estimate half-lives and assign mRNAs into different stability classes. Our data suggest that some mRNAs can be degraded in P-bodies, as the unstable mRNAs we have investigated are more highly colocalised with P-bodies and we can detect 3′ mRNA fragments in P-bodies. Overall, our data reveal the contribution of mRNA stability to shaping mRNA levels during early embryogenesis and provide insight into how mRNA stability is regulated.

## Results

### Isolation of mRNA from early embryos captures high-resolution transcriptional dynamics

To investigate mRNA accumulation dynamics during early *Drosophila* embryogenesis, we first generated a total RNA-seq time series. The early *Drosophila* embryo undergoes a series of 14 nuclear cycles within a common cytoplasm (nc1-14). RNA was isolated from single early *Drosophila* embryos at 10 time points, starting at nc11, approximately 90 minutes after egg lay (AEL) and prior to the onset of bulk zygotic transcription through to the beginning of gastrulation (Fig 1A). Embryos were collected from a His2AvRFP transgenic line and precisely staged at nc11, nc12, nc13, and nc14 by calculating an internuclear distance ratio (S1 Fig). Single embryos were collected in triplicate 5 minutes after the nc11 and nc12 divisions, both 5 and 15 minutes after the nc13 division, then at 15 minute intervals during the long nc14 interval, with the final time point corresponding to the appearance of the cephalic furrow (CF) (Fig 1A and Table 1). As male and female embryos have differences in X chromosome transcription due to dosage compensation [24], we used PCR to determine the sex of each embryo and select female

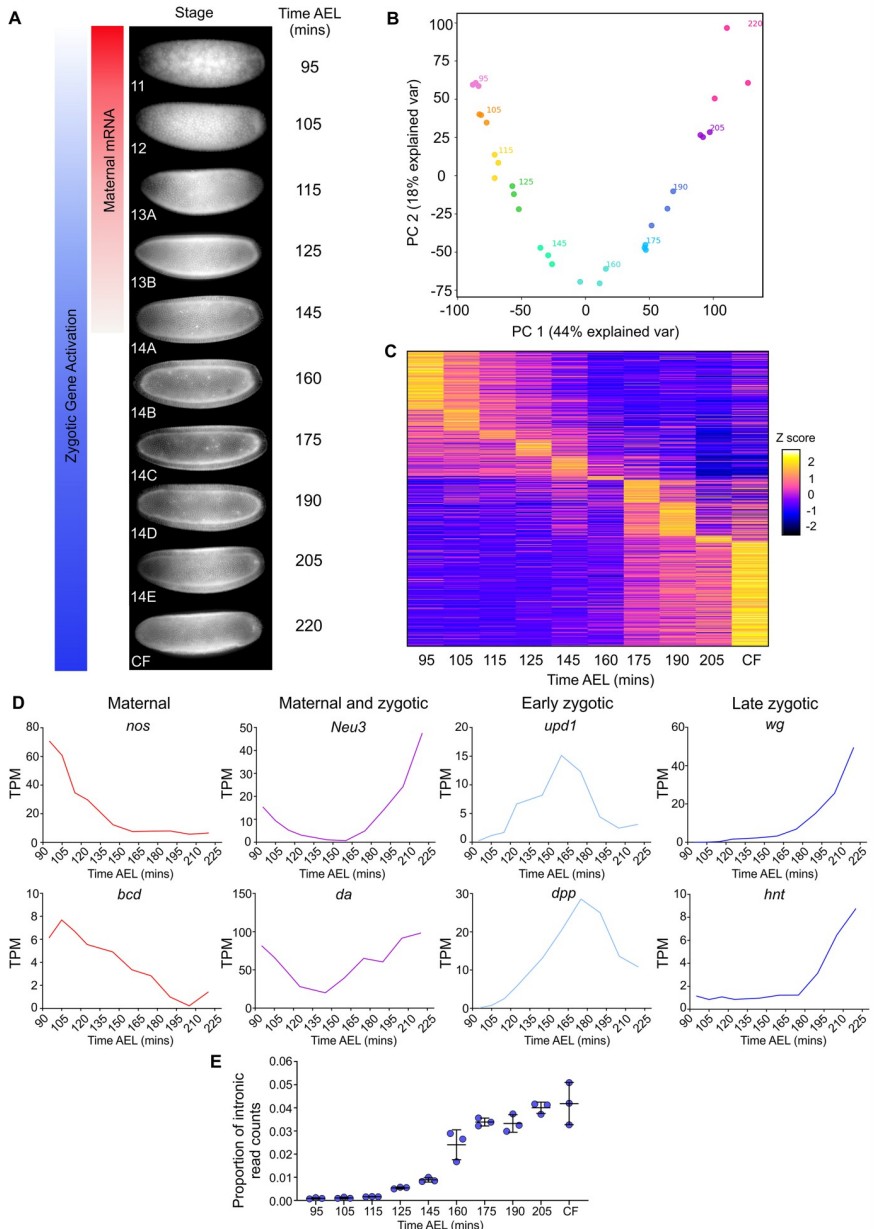

**Fig 1. Total RNA-seq captures dynamic gene expression during early *Drosophila* development.** (A) The time points used to collect RNA-seq samples, with approximate times AEL at 20°C, are shown ranging from early nc11 through to CF formation. The schematic highlights how the embryo switches from relying on maternally loaded RNAs (red) to activating its own zygotic transcription (blue). (B) Principal component plot of the RNA-seq samples shows tightly correlated replicates at each time point. (C) A heatmap of all expressed transcripts with TPM >1 (18,159 transcripts). Data are Z normalised and ordered along the y axis by the time point of peak expression. (D) Different examples of transcript dynamics captured by the data, as indicated by the graph titles. (E) Proportion of normalised intron reads (RPKM) throughout the time series. Underlying data are available at ArrayExpress (E-MTAB-11580), S1 Data, and S1 Table. AEL, after egg lay; CF, cephalic furrow; RPKM, reads per kilobase of transcript per million reads mapped; TPM, transcripts per million.

embryos for analysis. We sequenced total RNA following rRNA depletion, rather than selecting for polyadenylated RNA, allowing us to capture intronic reads and other noncoding RNA species. The intronic reads allow quantification of nascent, unspliced transcripts and also

**Table 1. *Drosophila* embryo samples collected for RNA-seq time course.**

| Sample collection time information | Sample stage ID | Approximate time AEL at 20˚C (mins) |
| --- | --- | --- |
| nc11 + 5 mins | nc11 | 95 |
| nc12 + 5 mins | nc12 | 105 |
| nc13 + 5 mins | nc13A | 115 |
| nc13 + 15 mins | nc13B | 125 |
| nc14 + 15 mins | nc14A | 145 |
| nc14 + 30 mins | nc14B | 160 |
| nc14 + 45 mins | nc14C | 175 |
| nc14 + 60 mins | nc14D | 190 |
| nc14 + 75 mins | nc14E | 205 |
| CF | CF | ≥220 |

AEL, after egg lay; CF, cephalic furrow.

detection of early zygotic expression by distinguishing zygotic transcripts containing introns from maternally loaded spliced mRNAs.

We detected a total of 18,159 transcripts during early embryogenesis representing 9,026 unique genes. Using principal component analysis (PCA), we observed that the first 2 principal components represented 44% and 18% of the variation, respectively, and the replicates at each time point clustered together (Fig 1B). This suggests the biological age of the embryos explains the majority of variation within the data rather than differences between replicates, indicating the high quality of the libraries. Transcript levels across embryogenesis were visualised as a heatmap, with the transcripts ordered based on the time point of peak expression (Fig 1C). We classified 4,897 early peaking transcripts (at the 95 or 105 minute time points) as maternal and 13,262 transcripts peaking after 105 minutes as zygotic. Of the zygotic transcripts, 23% show peak expression early in nc13 or the start of nc14 (between 115 and 160 minutes inclusive) and the remainder show late peak expression after 160 minutes. Analysis of different dynamically expressed genes showed that our dataset included well-characterised maternal (*nos* and *bcd*), maternal and zygotic (*Neu3* and *da*), early zygotic (*upd1* and *dpp*), and late zygotic (*wg* and *hnt*) mRNAs (Fig 1D).

As we sequenced total RNA, we determined the number of reads that mapped to introns as well as exons and transcripts (S1 Table). Analysis of the distribution of intronic reads shows an even read coverage across introns over all time points (S2A Fig). Only a very small proportion of transcripts at time points 105 to 125 minutes have intronic reads (Fig 1E), suggesting there is only minor zygotic transcription of intron-containing genes during these early stages. Previous studies have shown that the earliest zygotic activation of the *Drosophila* genome is biased towards expression of short intronless genes [25,26], which we cannot distinguish from maternally deposited transcripts at the early time points in our data. In addition, the early nuclear cycles are short, limiting the time period of active transcription. Nonetheless, 8 genes have detectable levels of intron signal at nc12 and nc13A, suggesting early zygotic transcription (S2C and S2D Fig).

The proportion of intronic reads increases significantly at 125 minutes (Figs 1E and S2E), then there is a further large increase around mid-nc14 (Fig 1E), when bulk activation of zygotic transcription occurs [24]. We detect 7,276 zygotically expressed genes, similar to a previous estimate based on GRO-seq data [27]. The benefit of the high temporal resolution of our data can be seen in examples of transient gene expression, such as the gene *runt* (*run*) that is expressed at nc12 then peaks in early nc13 (S2D Fig). *run* has essential roles in patterning and

transcriptional control of sex determination in early development, so the precise temporal regulation of its expression is likely to be important for these functions [28]. Additionally, we observe temporal changes in mRNA isoforms during development, exemplified by the genes *Meltrin* and *thickveins* (*tkv*) (S3A and S3B Fig). Their isoforms have altered coding sequences, which for the zygotically expressed isoform of the BMP receptor Tkv results in a shorter extracellular ligand-binding domain. We also detected expression of noncoding RNA species, such as those in the bithorax complex (S3C Fig). Overall, due to the high temporal resolution of our data and the ability to detect noncoding RNAs, we have a high-quality dataset to investigate transcriptional dynamics in early *Drosophila* development.

## Gaussian process regression provides estimates of transcript half-lives in early embryogenesis

As the degradation of maternal transcripts has been studied previously [17], we focused on the kinetics by which zygotic mRNAs are cleared in the early embryo. We used the intronic reads in our total RNA-seq dataset to represent pre-mRNA levels as a proxy for the transcription rate, while exonic reads reflect mature mRNA levels (Fig 2A). The intronic reads are correlated with NET-seq data from early embryos [29] ($\rho = 0.46$, $p = 5.1 \times 10^{-11}$) (S2B Fig), consistent with the intronic signal reflecting transcription dynamics. We assume that introns are co-transcriptionally spliced and rapidly degraded. In support of this, NET-seq data suggests that >95% of splicing events are co-transcriptional in the *Drosophila* embryo [29] and modelling of metabolic labelling and sequencing data from *Drosophila* S2 cells revealed that the median half-life of introns is 2 minutes [30].

We used a Gaussian process (GP) regression model [31,32] of mRNA accumulation and degradation to estimate zygotic transcript half-lives from the intronic and transcript expression RNA-seq data (Fig 2B). Before fitting the GP regression, we applied a dynamic filter where we computed a log-likelihood ratio test between 2 GP regression models: a dynamic model with a radial basis function (RBF) kernel and a noise model to obtain genes that are differentially expressed. We then strictly filtered the dynamic data to select 593 mRNAs, which are purely zygotically transcribed and have very low reads at the first time point (<0.5 TPM (transcripts per million)). From these, we filtered further to select transcripts with a correlation between the mRNA and pre-mRNA above 0.4. This is a mild correlation threshold that we applied to exclude transcripts for which pre-mRNA and mRNA are unrelated and it would therefore be hard to fit the model. As we filtered our data to select for zygotic mRNAs, we expect a positive correlation as gene introns and exons would increase together. The model uses a GP that specifies a prior distribution over possible underlying functions before observing the data. This nonparametric prior is governed by ordinary differential equations (ODEs), which describe the transcription regulation process. Once the data are observed, Bayesian inference is used to infer the posterior distribution. The posterior distribution allows quantifying uncertainty in the model as it reflects possible functions that can explain a given dataset. Credible regions are derived from the posterior distribution to quantify the uncertainty at 95% confidence level. The ODE describing the system is shown in Fig 2B from which the splicing and degradation rates which are inferred using the GP regression. We assume that introns are spliced at the same rate for each mRNA, consistent with evidence from S2 cells that introns from the same mRNA tend to have similar splicing rates [30]. We normalised the read counts by the intron lengths so that the *S* parameter has comparable meaning for each mRNA, but allow it to differ from mRNA-to-mRNA to account for variation in splicing rates across mRNAs [29]. For transcripts, we used TPM units that normalises for transcript length.

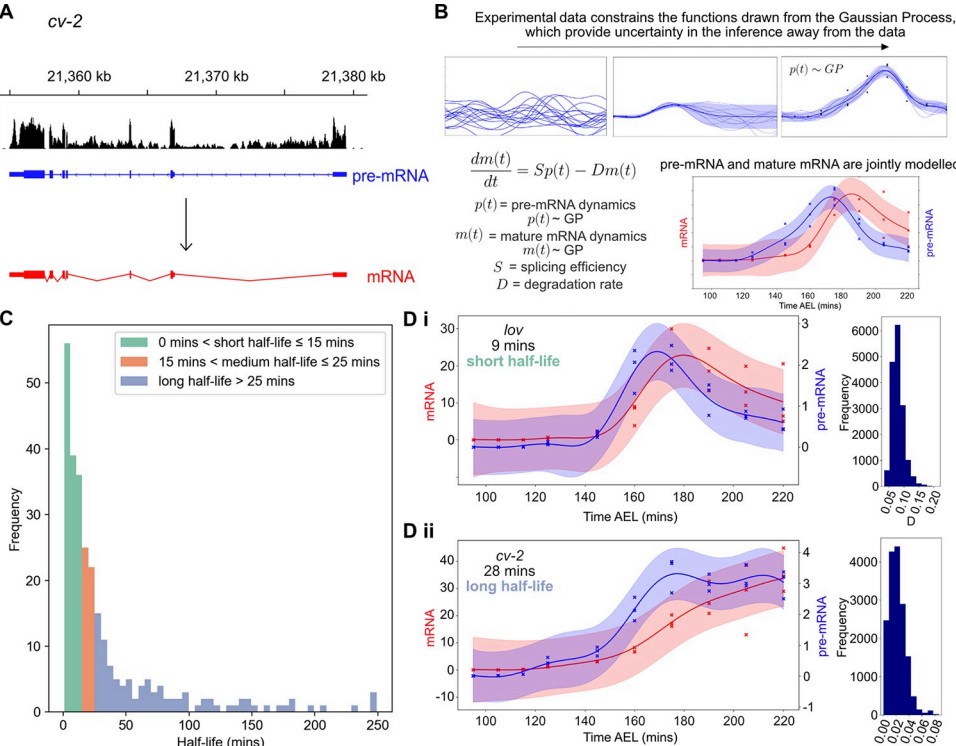

**Fig 2. GP regression provides estimates of transcript half-lives in early embryogenesis.** (A) Reads aligned to intronic and exonic regions of genes are used to represent pre-mRNA (blue) and mature mRNA levels (red), respectively. (B) Schematic of GP regression and the ODE model that shows the evolution of the mature mRNA dynamics is described by the pre-mRNA data over time, sculpted by the splicing (*S*) and degradation (*D*) parameters. Pre-mRNA and mature mRNA are therefore jointly modelled using GPs related by the ODE. (C) Half-life results for 263 transcripts estimated using the GP model. Transcripts are divided into short, medium, and long half-lives and coloured accordingly. (D) Examples of data for a short (Di) and a long (Dii) half-life mRNA, *lov* and *cv-2*, respectively, fit using the GP model. Pre-mRNA is shown in blue and mature mRNA in red, shaded areas represent credible regions and crosses mark the data for each experimental replicate at each time point. Posterior distributions for the degradation parameter *D* for each mRNA are shown to the right. Underlying data can be found in S1 Data, S1 and S2 Tables. GP, Gaussian process; ODE, ordinary differential equation.

The model provides half-life estimates for 263 zygotic transcripts corresponding to 186 genes (S2 Table). The distribution of these, coloured by short, medium, or long half-life, can be seen in Fig 2C, with the mean half-life at 35 minutes and median at 16 minutes. Fig 2D shows examples of an mRNA with a short (Di) and a long (Dii) half-life, estimated using the GP model. Parameters were determined for these mRNAs, along with associated uncertainty, using Markov chain Monte Carlo methods and the posterior distributions on the degradation rate *D* are displayed. *jim lovell* (*lov*) mRNA, a BMP target gene encoding a putative transcription factor [33,34], has a short half-life of 9 minutes, whereas the *crossveinless-2 (cv-2)* mRNA, encoding a secreted protein that binds BMPs [35], has a longer half-life of 28 minutes. Full parameter estimates and credible regions are shown in S4A Fig.

As the dynamic embryonic mRNAs are not at steady state, a previously described smFISH-based method developed in human cells [22] was unsuitable for validation of half-lives. An alternative method exploited the arrest of transcription during mitosis to calculate the *snail* mRNA half-life in the *Drosophila* embryo, based on quantitation of mRNA numbers before and after mitosis [23]. However, we found the variation between transcript numbers in different embryos to be greater than any reduction that would be expected over such a short time

frame (approximately 4 minutes) due to degradation (S4B–S4E Fig). As a result, any reduction due to degradation is masked by high variation between embryos, as has previously been observed for other mRNA numbers in the *Drosophila* embryo [36]. The *snail* mRNA numbers are tightly controlled by negative autoregulation [23], suggesting that *snail* may be uniquely suited to this method for calculating half-life.

In the absence of direct half-life validation, we determined whether the types of factors encoded by mRNAs with short and long half-lives have functions compatible with their inferred stabilities. Gene Ontology (GO) analysis reveals that, compared to all dynamic transcripts in the RNA-seq data, those encoding transcription factors and cell adhesion proteins are enriched in the short ($p$-values $2.3 \times 10^{-8}$ and $4.1 \times 10^{-5}$, respectively) but not long half-life mRNAs. Transcription factors have previously been reported to be encoded by unstable mRNAs [17,37,38]. Therefore, this approach has allowed the classification of transcripts into half-life categories where short half-life mRNAs are enriched for protein functions reflected by their stability.

## Clustering reveals how degradation shapes mature mRNA dynamics

We next addressed how post-transcriptional regulation contributes to the range of mature mRNA dynamics seen in our data, by combining clustering analysis with our modelling of transcript half-lives. The pre-mRNA data were clustered using GPclust (S5 Fig) [39]. From the intronic clusters, 6 highly populated intronic clusters that together exhibit a variety of interesting mRNA dynamics are shown in Fig 3Ai. The genes in each cluster share similar pre-mRNA profiles and therefore transcription dynamics. All of the pre-mRNAs in intronic cluster 5 were then sub-clustered based on their mature-mRNA profiles (Fig 3Aii), which revealed that a range of mature mRNA dynamics arises from this single transcriptional profile. The zygotic mRNA subclusters for intronic cluster 2 also display a range of mature mRNA dynamics and are shown in S6 Fig. The GP model sheds light on how these various dynamics arise, due to differences in the half-lives of transcripts in each cluster (Figs 3Aii and S6). It is clear that the pattern in the shape of the time series is reflected in the different half-lives of the clusters; clusters that have a stronger peak have a shorter half-life and higher degradation rate, whereas those which continually increase across the time period have a long half-life and low degradation rate.

As the clustering data indicated that half-life contributes to the shape of the mature mRNA profile, we further investigated the relationship between the relative timing of the peak of the pre-mRNA and mature mRNA. Visualisation of the gene-level pre-mRNA and mature mRNA data from the zygotic subset as heatmaps, reveals that for a given pre-mRNA peak time, there are a range of mature mRNA peak times with different delays (Fig 3B). Delay is defined as the time difference at which the peak is observed for the pre- and mature mRNA. The pre-mRNA and mature mRNA data were modelled using a GP that was then sampled with $n = 100$, so that the time delay between the peaks could be determined and the uncertainty in the estimate quantified (Fig 3C). The relationship between delay and half-life, for each transcript that has been modelled, is shown in Fig 3D. There is a moderately positive yet significant correlation between the 2 variables. Fig 3E shows the data as a confusion matrix in order to assess whether delay is predictive of half-life. Enrichment along the diagonal supports this; 63% of short delay mRNAs have short half-lives; 67% of medium delay mRNAs have medium half-lives, and 72% of long delay mRNAs have long half-lives. Together, these results reveal how post-transcriptional regulation is able to shape mature mRNA dynamics through regulation of mRNA half-lives and that the time delay between maximum expression of the pre-mRNA and mature mRNA can be used as an indicator of mRNA stability.

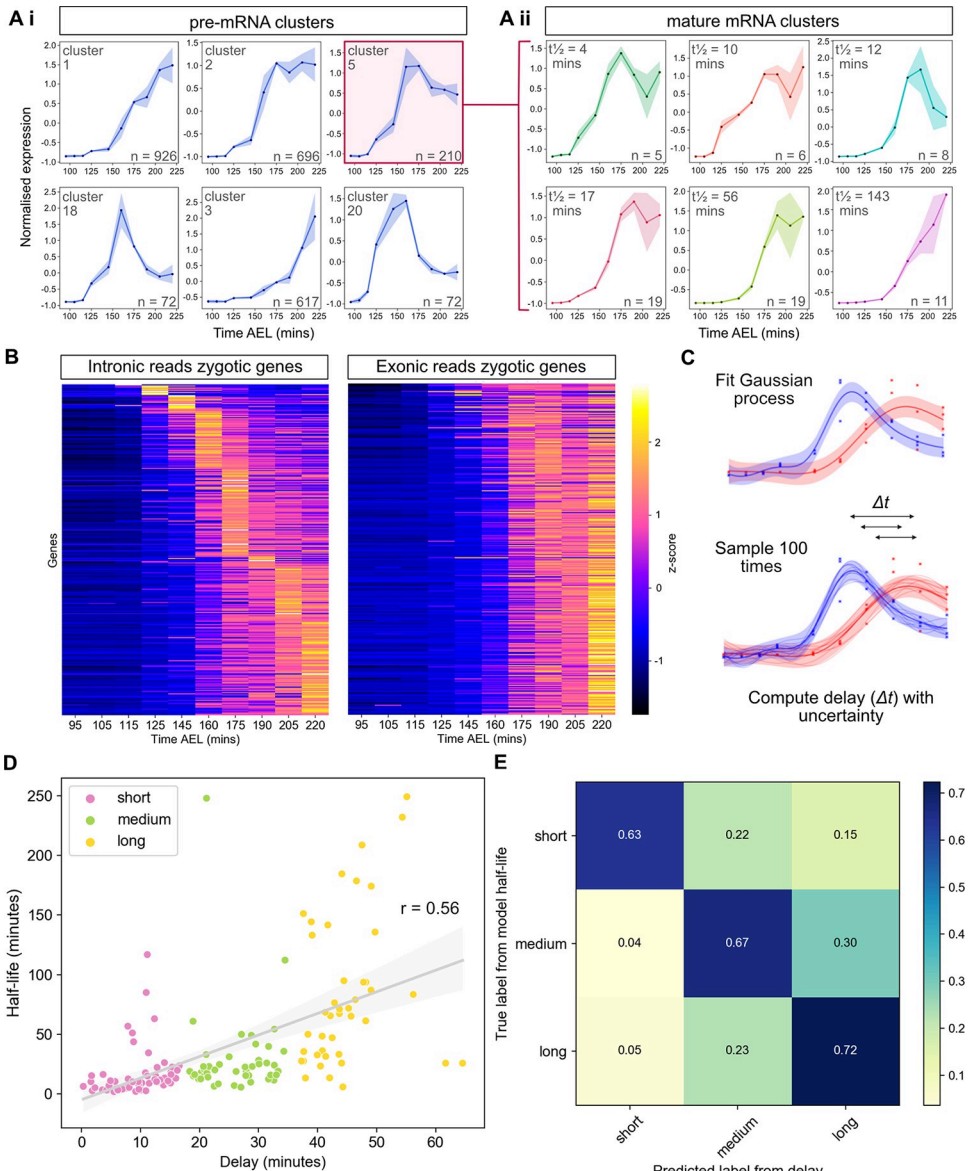

**Fig 3. Degradation regulates mRNA dynamics and can be approximated using the time delay between pre-mRNA and mature mRNA expression.** (A) Clustering of pre-mRNA (Ai) and mature (Aii) mRNA time series. Plots show the mean expression value at each time point for all mRNAs in a cluster and the shaded area shows the standard deviation. The intronic cluster ID (Ai) or mean half-life (Aii) is shown in the top left and the number of transcripts in each cluster in the bottom right. (B) Heatmaps of pre-mRNA and mRNA expression at the gene-level for the subset of zygotic transcripts, ordered by the time point of maximum pre-mRNA expression. (C) Schematic illustrating the estimation of temporal delays between pre-mRNA and mature mRNA peak times ($\Delta t$, arrows) by fitting and sampling from a GP. (D) Correlation of the delay between the peak of pre-mRNA and mature mRNA against inferred half-life (Pearson's r = 0.56, $p = 6.5 \times 10^{-14}$ for testing non-correlation). Points representing transcripts are coloured by time delay category. (E) Confusion matrix comparing mRNAs categorised into short, medium, and long delays and their respective half-life categories. Numbers in the boxes indicate the fraction of mRNAs with a given delay in the corresponding half-life category. Underlying data can be found in S1 Data and S4 Table. GP, Gaussian process.

## The short half-life mRNAs tested are more compact

The degradation of an mRNA in the cytoplasm can be closely linked to its translation [1]. We therefore investigated how mRNA half-lives are shaped by both structural and sequence

features known to influence translation. Regulatory sequences controlling mRNA degradation, translation, and localisation are frequently located in the 3′ UTR [40]. We found that 3′ UTR length does not have any significant correlation with our inferred half-lives (S7A Fig), in agreement with previous studies of mRNA stability in late stage *Drosophila* embryos [37]. Similarly, there is no relationship between transcript length and stability in our dataset (S7B Fig).

Due to the links between mRNA decay, translation efficiency, and codon optimality [41], we next investigated whether there is a relationship between half-life and both the translation efficiency and codon usage across the transcripts within our dataset. Using published ribosome footprint profiling data from 2- to 3-hour embryos [42], we plotted the translation efficiency and half-life for each of the transcripts within our dataset and observed no significant correlation between translation efficiency and half-life (Fig 4A). To extend this analysis, we also determined the codon stabilisation coefficient (CSC) for each codon that is a measure of the correlation between codon usage and stability of mRNAs. We plotted the CSC of each codon ordered by this value from highest to lowest (Fig 4B) and examined the identity of optimal codons previously defined in *Drosophila* embryos [43] and their occurrence within the CSC plot. The proportion of optimal codons is not significantly enriched within the positive and negative CSC groups (33% versus 39%, $p$ = 0.79, Fig 4B). There is also no significant difference in the proportion of optimal codons for transcripts within each of the different categories of half-life (Fig 4C) and clustering mRNAs based on codon usage showed that different clusters had similar half-lives (S7C Fig). Together, these data suggest that mRNA half-life does not depend on translation efficiency and that there is no relationship between optimal codon usage and half-life within our dataset.

We next used imaging to analyse mRNA compaction in the context of stability. A more open conformation has been detected for specific mRNAs when they are being translated [44–46], raising the possibility that a particular conformation may also influence mRNA stability. We therefore selected a set of 11 zygotic mRNAs, 4 each from the medium and long half-life categories and 3 from the short half-life category (S8A Fig). A fourth short half-life mRNA, *Neuroectoderm-expressed 2* (*Neu2*), is too short to separate the probe sets for compaction analysis but is included in further analysis (see later). We used dual-colour smFISH probes to visualise their 5′ and 3′ ends, and quantitate the distance between them, in fixed embryos (Fig 4Di). A representative smFISH image for one of the mRNAs, *lov*, is shown in Fig 4E, images for the other mRNAs tested are shown in S8B Fig.

For each image, the number and position of the 5′ and 3′ signals were collected and pairs were identified by solving a paired assignment problem (Fig 4Dii). For each pair, the distance between the 5′ and 3′ signals was then measured; only ends with a distance less than 300 nm were assigned as the same mRNA [46]. First, we estimated our smFISH detection efficiencies using alternating fluorophores for the *otd* and *lov* mRNAs (S9Bi Fig). These data reveal mean detection efficiencies of approximately 70% (S9Bii Fig), which is in the 70% to 90% range reported from other smFISH studies [47–51]. However, we note that our detection of the 670 labelled probe sets is generally slightly poorer than that of the 570 probes due to a lower signal to noise, consistent with findings from a previous study that used 670 labelled probes [49].

The distributions of end-to-end distances for each of the mRNAs tested reveal that short half-life mRNAs are significantly more compact, based on a smaller end-to-end distance, than mRNAs in the medium and long half-life categories (Fig 4F). Considering the lower detection limit of the imaging setup we used is approximately 120 nm (see Materials and methods), we found that for our alternating probe sets and the 5′ and 3′ compaction data, the *otd* and *lov* short half-life mRNAs had an end-to-end distance that is very compact and close to this limit (Fig 4F). No significant difference is observed in the end-to-end distance for mRNAs in the medium and long half-life categories; however, we did find some mRNAs in the long category

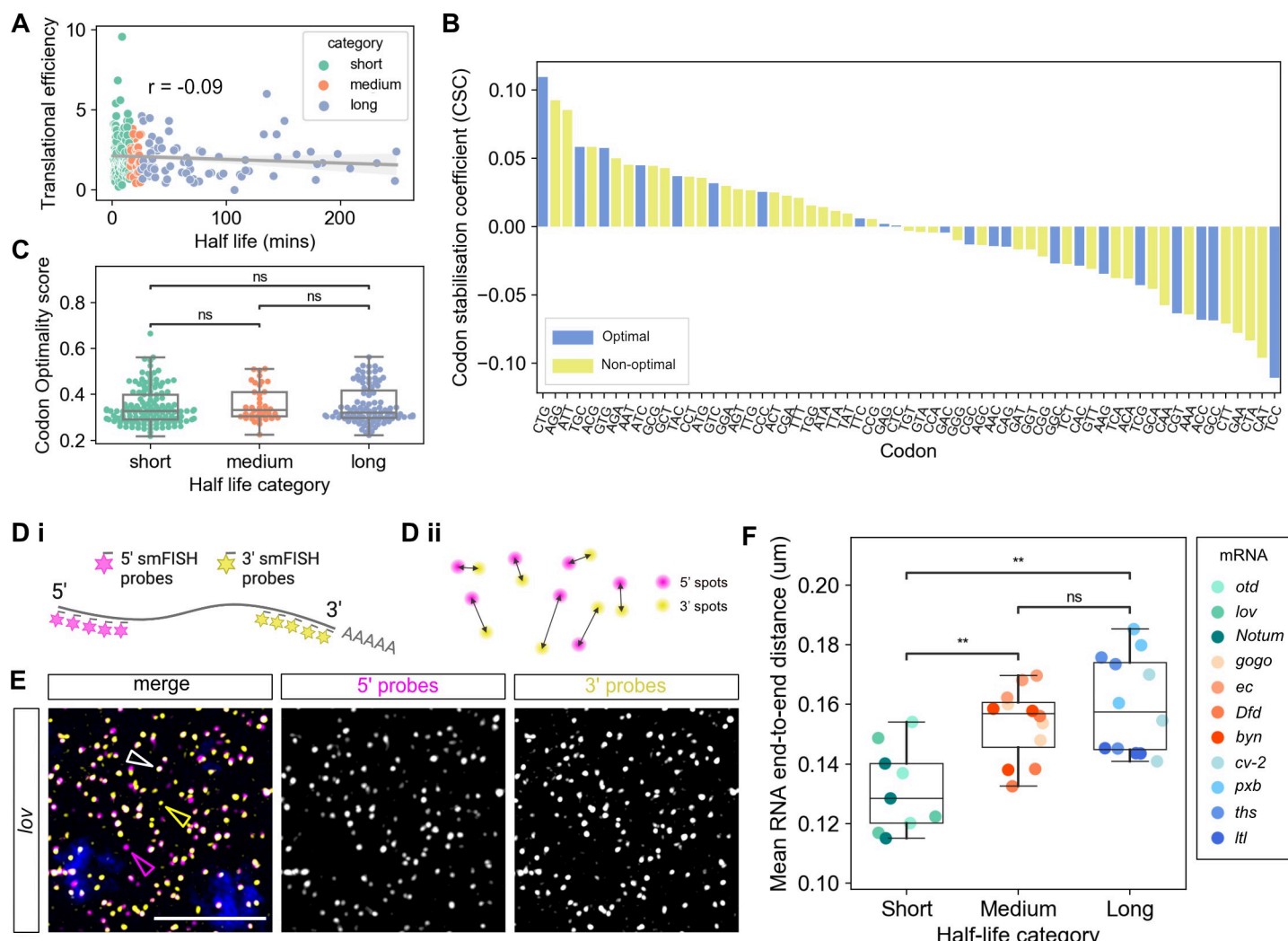

**Fig 4. mRNA properties and stability.** (A) Half-life (x-axis) versus the translation efficiency from 2–3 hour embryos [42]. In this and subsequent panels points representing transcripts are coloured by half-life category (Pearson's r = −0.09, *p* = 0.18). (B) CSCs calculated from our estimated half-lives showed no difference in the proportion of optimal (blue) and non-optimal codons (yellow) (chi-squared test *p* = 0.79). (C) Proportion of optimal codons within transcripts from each half-life category. No significant difference was observed in the proportion of optimal codons within each category, tested by independent *t* test (short vs. med *p* = 0.5; short vs. long *p* = 0.7; med vs. long *p* = 0.7). (Di) Schematic showing detection of the 5′ (magenta) and 3′ (yellow) ends of each mRNA with different smFISH probe sets. (Dii) Spots belonging to the same mRNA are matched (see Materials and methods). (E) Maximum projection of 6 slices from a confocal image showing smFISH detection of the 5′ and 3′ ends of *lov* mRNAs with lone 5′ ends, 3′ ends, and colocalised ends labelled by magenta, yellow, and white arrowheads, respectively. Scale bars: 5 μm. In the uncropped image from this embryo there are 5,668, 1,645, and 3,620 intact, lone 5′ and lone 3′ signals, respectively. For absolute numbers of intact mRNAs and lone ends for all mRNAs, see S9A Fig. (F) Graph shows the end-to-end distances of mRNAs with different stabilities, *n* = 3 embryos per mRNA. Data are mean distances across all colocalised mRNAs in each embryo (*n* > 220 whole RNAs for all images). mRNAs are grouped by their half-life category and the hue in each category corresponds to the order of the half-lives (lighter colour refers to shorter half-life). Short half-life mRNAs are more compact than both medium (*p* = 3.1 × 10$^{-3}$) and long (*p* = 1.6 × 10$^{-3}$) half-life mRNAs. No significant difference in the end-to-end distance was seen between medium and long half-life transcripts (*p* = 8.9 × 10$^{-1}$). Underlying data can be found in S1 Data. CSC, codon stabilisation coefficient; smFISH, single-molecule fluorescent in situ hybridisation.

were in a more open conformation than those in the medium category (Fig 4F). We also identified unpaired mRNA ends (see later), which were further apart than the 300 nm distance threshold used. Finally, quantitation of additional control smFISH experiments for some of the test mRNAs, in which the fluorophore dyes on each set of probes were switched (S9C Fig) in order to control for detection differences between the channels mentioned above, also revealed significantly shorter end-to-end distances for short half-life mRNAs (S9D Fig).

We find no correlation between compaction and mRNA length, as *brachyenteron* (*byn*) and *echinus* (*ec*) are the shortest and longest mRNAs tested, respectively. Taken together, these results suggest that within early *Drosophila* development, the decay of zygotically expressed genes is not strongly correlated with translation efficiency or codon optimality, but unstable mRNAs tend to be slightly more compact than medium and long half-life transcripts.

## Embryonic P-bodies are associated with unstable mRNAs and enriched in 3′ decay fragments

Cytoplasmic P-bodies have been implicated in mRNA degradation and translational repression in *Drosophila* [52]. Therefore, we investigated whether mRNAs with distinct stabilities are differentially localised to P-bodies. We visualised P-bodies using Me31B, a marker of P-bodies, including in *Drosophila* [53]. To detect Me31B, we used a fly stock carrying a GFP-Me31B protein trap with GFP inserted into the Me31B locus [54]. We quantified both single mRNAs using smFISH and P-bodies labelled by GFP-Me31B in fixed embryos. The same set of 11 mRNAs described above was used in these experiments with the addition of *Neu2*, a 1,126 nt mRNA that was unsuitable for compaction analysis due to its short length. Many GFP-Me31B foci were detected in the cytoplasm of early nc14 embryos (Figs 5A and S10A). These foci have a mean radius of 200 nm (S10B Fig), consistent with a previous observation that P-bodies in the embryo are smaller than those in the oocyte [55].

For each mRNA tested, a proportion of the individual mRNA signals colocalise with P-bodies (Figs 5A and S10A). As seen in Fig 5A, *orthodenticle* (*otd*) (also called *ocelliless*) mRNAs appear more highly colocalised with P-bodies than *larval translucida* (*ltl*) mRNAs. As *otd* has a much shorter half-life than *ltl* (3 minutes and 249 minutes, respectively), we examined whether this was a trend across the set of test mRNAs. To quantitate colocalisation, we used a colocalisation index that controls for variation in mRNA and P-body numbers between embryos (Fig 5B). This analysis reveals that both the short and medium half-life mRNAs tested are significantly more colocalised with P-bodies than the long half-life mRNAs tested (Fig 5C). While the mean colocalisation index value for short half-life mRNAs is higher than that of the medium half-life mRNAs tested, this difference is not significant, due to higher variance in the colocalisation index of short half-life mRNAs as *lov*, a short half-life mRNA, has a particularly low colocalisation index.

Given the difference in P-body colocalisation observed for some of the test mRNAs, we extended this analysis by using published Me31B RIP-seq data from the early *Drosophila* embryo [52]. This analysis reveals a relatively weak but significant negative correlation between Me31B interaction and mRNA half-life in 1- to 2-hour embryos (Fig 5D) and 2- to 3-hour embryos (S10Ci Fig). This negative correlation between the Me31B RIP-seq data and our model half-lives is no longer significant when the RIP-seq data from 3- to 4-hour embryos are used (S10Cii Fig), a later stage than we have imaged. These data are consistent with a previously reported negative correlation between Me31B binding and mRNA stability in the *Drosophila* embryo, when fold change in mRNA abundance was used as a proxy for mRNA stability [52]. Together, our imaging data and the negative correlation between RIP-seq interaction and mRNA half-life suggest that in the *Drosophila* embryo P-bodies may be sites of mRNA degradation for at least a subset of mRNAs.

In our dual-colour smFISH images, we observed a proportion of unpaired 5′ and 3′ mRNA ends suggestive of degradation intermediates (Fig 4E). We detect more lone ends when we use 5′ and 3′ *otd* compaction probes, compared to alternating probes (S9E Fig), providing further support that some of the lone signals are due to mRNA degradation, as detection with alternating probes is more resistant to loss of mRNA 5′ and 3′ sequences. In addition, due to the short

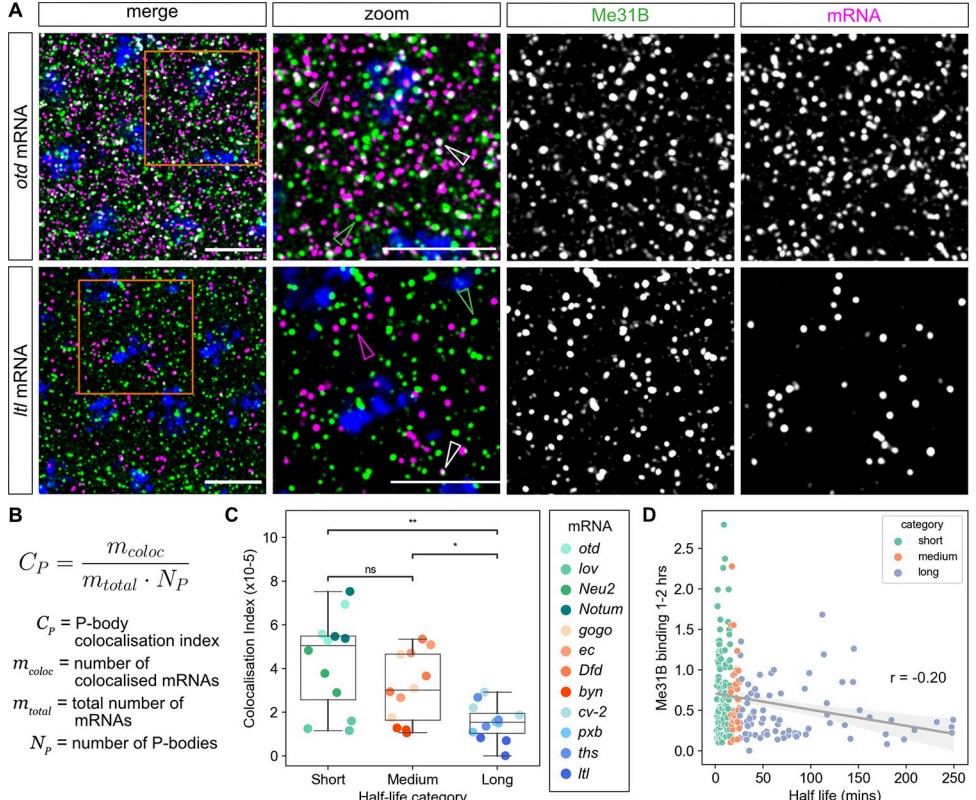

**Fig 5. Short half-life mRNAs are more colocalised with P-bodies in the early embryo.** (A) Confocal images of fixed, early nc14 embryos stained with smFISH probes for the indicated mRNAs (magenta) and labelled GFP-Me31B P-bodies (green). Scale bars: 5 μm. Images are maximum projections of 7 slices, with higher magnification images of the highlighted regions (orange box) shown. Individual mRNAs (magenta arrowheads), P-bodies (green arrowheads), and colocalised mRNA and P-body signals (white arrowheads) are highlighted. (B) The P-body colocalisation index used to calculate the normalised proportion of colocalised mRNAs, facilitating comparison between different mRNAs. (C) Graph of the P-body colocalisation index for the indicated mRNAs in early nc14. mRNAs are grouped by half-life category, within which they are coloured by half-life as per Fig 4F with points representing individual embryos. Both short and medium half-life mRNAs are significantly more colocalised with P-bodies than long half-life mRNAs ($p = 1.257 \times 10^{-3}$ and $p = 1.640 \times 10^{-2}$, respectively). Short half-life mRNAs are not significantly more enriched in P-bodies than medium half-life mRNAs ($p = 4.168 \times 10^{-1}$). (D) Half-life versus Me31B binding data from 1–2-hour embryos [52] (Pearson's r = −0.20, $p = 1.5 \times 10^{-3}$). Underlying data can be found in S1 Data. smFISH, single-molecule fluorescent in situ hybridisation.

length of *Drosophila* mRNAs, we are using 24 to 30 probes in each detection set. Therefore, it is likely that loss of binding of only a small number of probes from the 5′ or 3′ set is enough to take the signal below the detection threshold, facilitating our detection of partly degraded mRNAs.

In order to determine if these 5′ and 3′ fragments co-localised with P-bodies, we assessed whether the 5′ and 3′ probe sets colocalised with the GFP-Me31B P-body marker. An image of an early nc14 embryo is shown for the *Deformed* (*Dfd*) mRNA in Fig 6A, revealing that some complete mRNAs (orange arrowhead) and lone 3′ ends (yellow arrowhead) are colocalised with the P-body marker Me31B. However, colocalisation of lone 5′ ends with Me31B is less evident. For clarity, an equivalent region of an early nc14 embryo is shown as 3 colour images with only either the 5′ or 3′ end of *Dfd* mRNAs, Me31B, and DAPI (Fig 6B, data for the other mRNAs tested are in S11 Fig). For the analysis, we identified unpaired 5′ and 3′ ends as described above and assessed if there is an enrichment of either end in P-bodies (S12A Fig). In

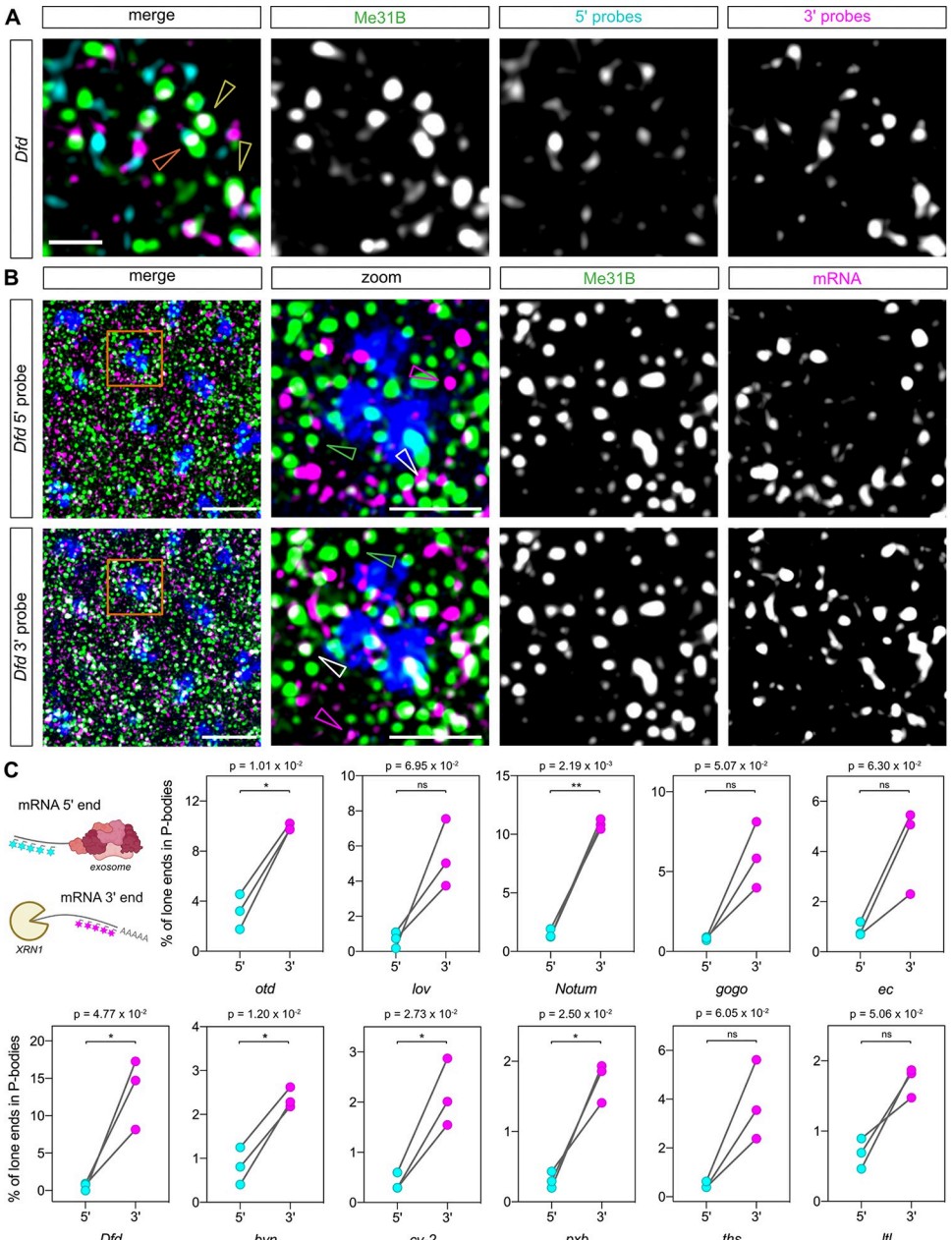

**Fig 6. mRNA 3′ end fragments are more associated with P-bodies than 5′ ends.** (A) Maximum projection (2 slices) of a confocal image showing smFISH staining of an early nc14 embryo with probes for *Dfd* 5′ (cyan) and 3′ (magenta) ends, GFP-Me31B (green). Examples where both the 5′ and 3′ ends or only the 3′ end is colocalised with P-bodies are indicated by orange and yellow arrowheads, respectively. Single channels for the smFISH and GFP-Me31B images are shown with the merged image. Scale bar: 1 μm. (B) As in (A) except the images (7 Z slices) show only 1 mRNA end (5′ in the top panels, 3′ in the lower panels) at a time for clarity. The mRNAs, GFP-Me31B, and DAPI are shown in magenta, green, and blue, respectively. A higher magnification image is shown as a merge and single channels, with an individual mRNA end (magenta arrowhead), P-body (green arrowhead), and colocalised mRNA end and P-body signals (white arrowhead) highlighted. Scale bars: 5 μm in merge and 2 μm in the higher magnification image. (C) Top left: Schematic showing how degradation by either pathway would result in a single fluorophore signal being detected. Quantification of the percentage of unpaired mRNA 5′ and 3′ ends colocalised with P-bodies relative to the total number of lone 5′ or 3′ ends; *n* = 3 embryos, paired *t* test used to determine significance with α = 0.05. For absolute numbers of intact mRNAs and lone ends, see S9A and S9C Fig. Underlying data can be found in S1 Data. smFISH, single-molecule fluorescent in situ hybridisation.

general, we do not see an excess of lone 3′ ends compared to 5′ ends across the mRNAs we tested (S9A and S9C Fig). However, quantitation of the proportion of single 5′ and 3′ signals that localise to P-bodies reveals a general trend of more unpaired 3′ ends in P-bodies, which is significant for over half the mRNAs investigated (Fig 6C). Similar results are obtained when the fluorophores on the *otd*, *Dfd*, and *cv-2* 5′ and 3′ probes are reversed (S12Bi Fig). Furthermore, this trend is lost when we use alternating probes for *otd* (S12Bii Fig). Taken together, these results suggest that the lone 3′ signals detected in P-bodies are consistent with them being 5′ to 3′ mRNA decay intermediates. Additionally, comparison of the proportion of lone ends versus intact mRNAs in P-bodies (relative to the total number of each) reveals that in general the proportion of intact mRNAs and lone 3′ ends in P-bodies is similar but there are more intact mRNAs than lone 5′ ends (S12C Fig). This detection of intact mRNAs in P-bodies may support a storage role in addition to 5′ to 3′ decay (see Discussion). Together, these data suggest that P-bodies are associated with mRNA degradation for at least some mRNAs in the early *Drosophila* embryo.

## Discussion

Here, using total RNA-seq time series data and GP regression, half-lives of approximately 260 mRNAs in early *Drosophila* development were derived. Our data support widespread post-transcriptional regulation of gene expression in early development, as we show that shared transcription profiles give rise to a range of mature mRNA dynamics due to differences in degradation. The RNA-seq time series that we have generated is high resolution with additional time points and over an extended period of early embryogenesis compared to published datasets [24,56]. In addition, our libraries are total RNA-seq rather than poly(A) selected, facilitating detection of noncoding RNAs and unstable RNA species, such as co-transcriptionally spliced introns. Our RNA-seq data reveal how expression of different mRNA isoforms for a given gene varies across early embryogenesis and we highlight examples where isoform changes alter the protein sequence of specific domains, potentially impacting on function.

A major advantage of our approach for estimating half-lives is that it does not require transcription inhibition that can affect mRNA stability or mRNA labelling that can be difficult to achieve in vivo and the labelling time can influence the estimates [19,20,57]. A different method that uses RNA-seq data to estimate mRNA half-lives has been described previously, which solves ODEs describing the RNA life cycle by adopting constraints on RNA kinetic rates [58]. An advantage of our approach is that, as GP regression is nonparametric, there is greater flexibility and sensitivity in the model to more accurately represent the variety of dynamics observed in the data. Additionally, it allows us to quantify uncertainty due to biological variation and measurement noise. Using Bayesian inference, we obtain posterior distributions for degradation parameters and thus quantify uncertainty about their possible values.

Disadvantages of our approach are firstly that it is not global as approximately 20% of *Drosophila* genes expressed in our dataset do not contain introns. Secondly, even for mRNAs with introns, we only derived a proportion of transcript half-lives from the dataset due to strict filtering to ensure that there is signal in both the intron and transcript expression time-series meaning genes with small introns and therefore poor signal would also be excluded. Thirdly, for mRNAs with high degradation rates, there may be high uncertainty in the inferred degradation rate since the splicing rate and degradation rate estimates become difficult to disentangle (for simulations demonstrating the reach of the model, see S1 Supplementary methods). Fourthly, the modelling requires some computational expertise in this area to implement on a new dataset. Potential solutions to overcoming these issues would be to generate a Pol II ChIP-seq time series for the transcription profiles that would allow intronless genes to be studied.

Looser filtering could be applied to provide half-lives for more mRNAs, although this would potentially lower confidence in the estimates. Finally, the delay between the peak of the pre-mRNA and mature mRNA could be measured as a simpler approach for categorising stability, as we have shown that the stability of a transcript can be classified using this delay.

The half-lives we estimated for approximately 260 zygotic transcripts in the early embryo have a median of 16 minutes. Previous half-life estimates of 7 to 14, 13 and 60 minutes described for the zygotic *fushi tarazu*, *snail*, and *hunchback* mRNAs, respectively, in the early *Drosophila* embryo [23,59,60] fall within the range of half-lives we observe. Moreover, the wide range of half-lives we estimate in the embryo suggests that mRNA stability is an important checkpoint in the regulation of gene expression. The median half-life we estimate is shorter than that of 73 minutes calculated for older (stage 12 to 15) *Drosophila* embryos, in a study that used a 4-hour pulse-chase labelling [37]. While the pulse-labelling timing may skew some of the half-life estimates [57], the shorter median half-life in the early embryo may reflect its rapid initial development. Early embryogenesis is characterised by short mitotic cycles [61] and fast rates of transcription [62] and translation [63], with the resulting localised gene expression patterns specifying 3 tissues along the dorsal–ventral axis in a time period of only 90 minutes [64]. Therefore, mRNA degradation rates may be faster than at other stages to limit the perdurance of transcripts encoding factors affecting cell fate.

GO analysis revealed an enrichment among the short half-life mRNAs for those encoding transcription factors and cell adhesion proteins. This is consistent with transient localised expression of key transcription factors in the early embryo and the mRNAs encoding transcription factors commonly being unstable [17,37,38]. Future studies will be able to determine how particular mRNA half-lives contribute to patterning by exploiting the extensive characterisation of gene regulatory networks in the early *Drosophila* embryo [65].

Previous studies have shown that mRNAs exist in a more open conformation during translation, while untranslated mRNAs are more compact [44–46] regardless of whether they are stress granule associated [44,45]. We found a trend that the 5′ and 3′ ends are closer for shorter half-life mRNAs. A more compact structure may facilitate degradation as 5′ to 3′ decay involves communication between deadenylation and decapping factors [66]. Alternatively, the shorter distance between 5′ and 3′ ends could reflect a transient interaction associated with degradation, which our smFISH snapshot images capture more frequently for the less stable mRNAs.

Codon identity and translation efficiency have previously been shown to be an important determinant of mRNA stability in bacteria, yeast, *Drosophila*, zebrafish, and mammalian cells [41]. Optimal codons, which are determined by codon bias in abundant mRNAs and the gene copy number of their cognate tRNA, lead to efficient translation and are enriched in stable transcripts [41]. However, our data suggest that codon optimality and translation efficiency are not major determinants of mRNA stability for early zygotic transcripts. A correlation between codon optimality and mRNA stability was observed for maternal mRNAs during the maternal to zygotic transition in the early *Drosophila* embryo, which likely contributes to clearance of maternal transcripts [67]. Optimal codons are also associated with stable mRNAs in late-stage *Drosophila* embryos, but not in neural tissues, potentially because mRNA stability regulation by RNA-binding proteins dominates in the nervous system [43]. The effect of codon optimality may also be masked for early spatially regulated zygotic transcripts. This could be due to additional regulation by RNA-binding proteins and miRNAs [5], a dependence on a particular distribution of non-optimal codons for instability [68] and/or tRNA abundance being a poor proxy for aminoacylated tRNA levels for a subset of tRNAs. In support of the latter, low aminoacylation of particular tRNAs has been observed in the mouse liver that may contribute to inefficient translation [69].

Our imaging data on the test set of mRNAs show that those with short half-lives tend to be more colocalised with the P-body marker Me31B than more stable mRNAs in the early embryo. Consistent with this, using published Me31B RIP-seq data from the early *Drosophila* embryo, we find a significant correlation between Me31B interaction and mRNA half-life across the set of approximately 260 mRNAs for which we estimated half-lives. The stronger association of short half-life mRNAs with P-bodies and our ability to detect mRNAs lacking their 5′ end in P-bodies suggests that 5′ to 3′ mRNA decay can occur in P-bodies in the early *Drosophila* embryo. However, the majority of the lone 3′ ends we detect are in the cytosol, suggesting that mRNAs can also undergo 5′ to 3′ decay outside of P-bodies. We also note that the localisation of mRNAs with P-bodies is variable as, within the short and medium half-life categories, the *lov* and *byn* mRNAs are less colocalised with P-bodies. This suggests that for these mRNAs in particular, degradation in P-bodies may only have a minor contribution to their turnover.

Although we generally detect similar proportions of 5′ and 3′ end fragments of a particular mRNA in the cytoplasm, there is weaker colocalisation of 5′ end fragments with P-bodies. This observation suggests that 3′ to 5′ mRNA degradation by the exosome does not occur in P-bodies, consistent with components of the exosome being largely absent [4]. We also detect a similar proportion of intact mRNAs in P-bodies (relative to the total number in the cytoplasm), as we find for lone 3′ ends. The presence of intact mRNAs in P-bodies may reflect an mRNA storage role. Therefore, we speculate that in the *Drosophila* embryo mRNAs enter P-bodies where they can undergo either: (1) 5′ to 3′ degradation (hence the lone 3′ ends detected); or (2) transient storage before exit back into the cytoplasm for translation.

A role for P-bodies in 5′ to 3′ decay is consistent with early studies in yeast following the discovery of P-bodies [70] and with later work in *Drosophila* suggesting that Me31B is involved in mRNA degradation in the embryo following zygotic genome activation [52] and P-bodies are sites of mRNA degradation in intestinal stem cells [71]. In addition, the Xrn1 exonuclease localises to P-bodies in yeast, *Drosophila* and mammalian cells [72]. However, P-bodies have been implicated in mRNA storage and translational repression in mature *Drosophila* oocytes [55] and Me31B represses translation of maternal mRNAs in *Drosophila* embryos prior to zygotic genome activation [52]. Moreover, many lines of evidence from other systems argue against a role for P-bodies in mRNA degradation. These include an absence of detectable mRNA decay intermediates either following purification of P-bodies [73] or based on a live imaging approach [47], mRNA degradation when P-body formation is disrupted [73] and the ability of P-body mRNAs to re-enter translation [74,75]. Although the sequencing data following P-body purification from human tissue culture cells provided evidence for mRNA storage, and do not support a role for P-bodies in bulk mRNA degradation [73], we note that 2 pieces of data are potentially consistent with some degradation occurring in P-bodies. Firstly, there is a weak correlation between mRNA P-body enrichment and half-life, and secondly, a 3-fold difference in the median half-lives of the most strongly enriched versus depleted P-body mRNAs was observed [73].

We speculate that P-bodies are involved in both storage and degradation in an mRNA dependent manner, with features of an individual mRNA as well as the proteins present in P-bodies at a particular developmental time influencing which function dominates. In support of this, it is known that there are changes in P-bodies during *Drosophila* development, for example, from being large and viscous in the oocyte to smaller, more dynamic structures in the early embryo [55]. Moreover, at the maternal-to-zygotic transition some P-body proteins are degraded, including the Cup translational repressor protein, which may increase the prevalence of mRNA decay in P-bodies [52]. Our data suggest that the degradation of mRNAs in P-bodies contributes to the post-transcriptional regulation of zygotic mRNAs in the embryo.

Future studies exploiting the method developed for determining the protein and RNA contents of purified P-bodies [73], along with the power of *Drosophila* genetics and single-molecule imaging, will reveal how P-bodies impact on mRNA stability or storage and cell fate decisions during development.

## Materials and methods

### Biological methods

**Fly stocks.** All stocks were raised at 25˚C and maintained at 20˚C for experiments on standard fly food media (yeast 50 g/L, glucose 78 g/L, maize 72 g/L, agar 8 g/L, 10% nipagen in EtOH 27 mL/L, and propionic acid 3 mL/L). The following fly lines were used in this study, $y^1$ $w^*$ (BDSC Stock #6599), $y^1$ $w^*$; P{His2Av-mRFP1}II.2 (BDSC Stock #23651) and $y^1$ $w^*$; P{w[+mC] = PTT-GB}me31B[CB05282] (BDSC Stock #51530).

**Staging and collection of embryos for RNA-seq.** Flies carrying His2AvRFP were allowed to lay on apple juice agar plates in small cages for 1 hour. Embryos were dechorinated in 50% bleach (2.5% final concentration of sodium hypochlorite diluted in distilled water) for 3 minutes and washed thoroughly in distilled water. Individual embryos were carefully transferred into a droplet of halocarbon oil (Sigma-Aldrich; a mix of 700 and 27 oil at a ratio of 1:4) on a rectangular coverslip (Deltalab, 24 × 50 mm, Nr. 1) and inverted over a cavity slide (Karl Hecht). Embryos were visualised and imaged with a Leica optigrid microscope at 20× magnification using a Texas red filter. Embryos were timed following the observation of a nuclear division, an image was taken and the embryo was immediately picked out of the oil droplet with a pipette tip and transferred to Eppendorf tubes containing 50 μL TRIzol Reagent (Invitrogen). Single embryos were crushed and homogenised using a pipette tip and an additional 450 uL Trizol added. Samples were immediately snap frozen in liquid nitrogen and stored at −80˚C until processing for nucleic acid extraction.

Ten time points were collected spanning early *Drosophila* embryonic development from nc11 through to CF formation (Table 1). Embryos were collected 5 minutes after nuclear division for nc11 and nc12, 5 and 15 minutes following the nc13 nuclear division, and every 15 minutes following the nc14 nuclear division as well as embryos that showed clear CF formation. This yielded samples covering every 10 to 15 minutes through development from nc11 to CF formation. The internuclear distance of 15 to 20 nuclei pairs per embryo was measured in Fiji and normalised to the whole embryo length to obtain an average internuclear distance per embryo (S1A and S1B Fig). This was compared to the internuclear distance of embryos of known stages to accurately confirm the nuclear cleavage stage and age of embryos. All embryos were collected at 20˚C with approximate time after egg lay in minutes shown in Table 1.

**Nucleic acid extraction and embryo genotyping.** Samples stored in Trizol (Invitrogen) were used for RNA and DNA extraction performed according to the manufacturer's protocol and resuspended in 10 μL (RNA) or 20 μL (DNA) nuclease free water. Extracted DNA was PCR amplified to sex the embryos by using Y chromosome specific primers to a region of the male fertility factor gene *kl-5*, forward primer 5′ GCTGCCGAGCGACAGAAAATAATGACT 3′ and reverse primer 5′ CAACGATCTGTGAGTGGCGTGATTACA 3′ [24] and control primers to a region on chromosome 2R forward primer 5′ TCCCAATCCAATCCAACCCA 3′ and reverse primer 5′ CCTACCCACAAGCAACAACC 3′. PCR reactions were performed in triplicate.

Total RNA was treated with TURBO DNA-free Kit Dnase (Invitrogen) and depleted of rRNA using the Ribo-Zero Magnetic Kit HMN/Mouse/Rat 24 Rxn (Illumina; Cat# MRZH11124) according to the manufacturer's low input protocol with 2 to 4 μL rRNA removal solution yielding a 20 μL final sample volume added to 90 μL magnetic beads. Beads

were resuspended in 35 μL resuspension solution and ribo-depleted total RNA was ethanol precipitated and resuspended in 18 μL FPF mix prior to RNA-seq library preparation.

**RNA-seq library preparation and sequencing.**   Three female embryos from each time point were used as replicates to make 30 individual RNA-seq libraries. Individual total RNA-seq libraries were prepared from ribo-depleted RNA using a TruSeq stranded library prep kit (Illumina) according to the manufacturer's protocol. Unique dual index adaptors were used for each library and they were pooled in equimolar concentration and run across 8 lanes on the flow cell of the HiSeq 4000 to obtain paired-end sequence reads. The average number of reads obtained per library was 105 million reads.

**Embryo fixation and smFISH.**   Flies were allowed to lay on apple juice agar plates in small cages for 2 hours at 25°C. After ageing for another 2 hours, 2- to 4-hour-old embryos were dechorinated in 50% bleach for 3 minutes and washed thoroughly in distilled water. Embryos were fixed as previously described [76] and stored in methanol at −20°C until required. Fixed embryos were placed in Wheaton vials (Sigma, Z188700-1PAK) for the smFISH reaction as described previously [77]. mRNA targets were detected in embryos using smiFISH probes designed to exonic sequences with 5′ end X flap sequences [78] and using secondary detection probes labelled with Quasar 570 or Quasar 670 fluorophore dyes (LGC Biosearch Technologies). Probe sequences are listed in S3 Table. DAPI (500 μg/ml) was added to the third of the final 4 washes of the protocol at a concentration of 1:1,000 and embryos were mounted onto slides in Prolong Diamond to set overnight before imaging. To visualise the membrane to age the embryos, a mouse α-Spectrin antibody (DSHB, 3A9 (323 or M10-2)) with an Alexa Fluor 647 Donkey anti-Mouse IgG (H+L) Highly Cross-Adsorbed Secondary Antibody (Thermo Fisher Scientific, A-31571) was used or a brightfield image was taken.

For compaction experiments, at least 24 probes were designed to each end of the mRNA (5′ and 3′) separated by at least 1.3 kb. As a control, fluorophore dyes were switched and the images from stained embryos analysed and quantified. Additional controls for *otd* and *lov* used adjacent probes with alternating Quasar dyes to determine the precision of detection of single mRNAs.

**Confocal microscopy of fixed embryos.**   A Leica TCS SP8 gSTED confocal was used to acquire images of the transcription sites (TSs), single mRNAs, and P-bodies within cells of fixed embryos using a 100×/1.3 HC PI Apo Cs2 objective with 3× line accumulation and 3× zoom for compaction and P-body colocalisation experiments, and 2× zoom for quantifying mRNAs for the half-life validation. Confocal settings were approximately 0.6 airy unit pinhole, 400 Hz scan speed with bidirectional line scanning, and a format of 2,048 × 2,048 or 4,096 × 4,096 pixels. Laser detection settings were collected as follows: PMT DAPI excitation at 405 nm (collection: 417 to 474 nm); Hybrid Detectors: Alexa Fluor 488 excitation at 490 nm (collection: 498 to 548 nm), Quasar 570 excitation at 548 nm (collection: 558 to 640 nm), and Quasar 670 excitation at 647 nm (collection: 657 to 779 nm) with 1 to 6 ns gating.

All images were collected sequentially and optical stacks were acquired at system optimised spacing. Imaging of the membrane using brightfield or anti-Spectrin antibody at the mid-sagittal plane of the embryo with 40× objective at 0.75× zoom and 1,024 × 1,024 format was used to measure the average length of membrane invagination from at least 5 cells. These measurements were used to select embryos of a similar age in early nuclear cycle 14 (10 μm membrane invagination). For all analyses, 3 separate embryos were imaged and quantified as independent replicates. smFISH images were deconvolved using Huygens professional deconvolution software by SVI (Scientific Volume Imaging). By deconvolving images taken on a Leica SP8 confocal, we estimate that our lower detection limit is approximately 120 nm.

**Image analysis.**   The spot detection algorithm Airlocalize [79] was used to detect and quantify TSs, single mRNAs, and P-bodies within confocal microscopy images. This software

fits a 3D Gaussian mask, a lower threshold is defined to separate signal from background, and the X, Y, and Z coordinates of spots and their intensities are output. Z stack images were first subsetted to detect TSs within the range of Z slices around the nuclei. Images were then processed again to detect single mRNAs in the full image. The TS data was then used to remove these high intensity spots from the single mRNA data. Detection of 5′ and 3′ single mRNA ends and P-bodies was performed separately on each corresponding channel image as appropriate.

**Half-life validation.**   For validation of half-lives as previously described [23], embryos were imaged at various time points during the 13th nuclear division (S4 Fig) using the DAPI channel and reference movies of His-RFP [77] to carefully stage the images. Single mRNAs were quantified using Airlocalize and the number per cell was calculated by dividing by the total number of pre-division cells in the images. The counts per cell were fitted with an exponential function, from which the half-life was determined. The signal to noise ratio in the data was then calculated from the change in the mean over the time course, divided by the average variance in mRNA numbers at each time point with sufficient data.

## Computational methods

**RNA-seq data processing and data filtering.**   The RNA-seq data were processed at the transcript level by alignment-free methods using Kallisto [80] and the *Drosophila melanogaster* Ensembl v96 reference transcriptome to infer transcript expression levels for modelling mature mRNA dynamics. Transcripts were filtered to remove any that had <1 TPM across the time course yielding 18,159 remaining transcripts. The transcript-level reads were then filtered for dynamic transcripts using GPcounts [81]. This approach fits the data for each gene using a GP with a dynamic kernel, and a GP with white noise kernel that exhibits no variation over time. The transcripts where the dynamic kernel is a better fit, measured via likelihood ratio test, are then extracted. For the transcript data, this reduced the numbers of mRNAs from 18,159 to 8,791.

The whole-embryo total RNA-seq dataset was also processed at the gene level in order to quantify the intronic reads, by aligning data to BDGP6 (dm6) using STAR with default parameters. FeatureCounts was used to get the counts data for exons and introns, respectively. Modified RPKM (reads per kilobase of transcript per million reads mapped) normalisation was applied to exon and intron counts data, where the total mapped reads for each library was used to address the sequencing depth for exon and intron counts from the same sample yielding 11,587 genes with a detectable level of expression (RPKM > 0).

To model the pre-mRNA dynamics, any genes without introns, or with zero intronic reads across all time points were removed to give a set of 5,035 genes and the intron sizes were then used to obtain length-normalised reads. The intronic read counts are divided by the total intron length for each gene and the exonic expression level is similarly normalised. For modelling the mature mRNA dynamics, the transcript-level alignment was used. A set of strictly zygotic transcripts was extracted from the dynamic dataset ($n = 8,791$) by filtering for transcripts with TPM < 0.5 at the first time point (t = 95) to give a set of 593 zygotic transcripts that were used in subsequent analysis. For the GP model, transcripts were subjected to a further filtering step where the correlation between pre-mRNA and mRNA was computed to extract transcripts where the correlation was above 0.4. For more details on filtering, see S1 Supplementary methods. For full time series data along with filtered subsets, see S1 Table.

**Analysis of intronic read coverage.**   BAM files were used to calculate the read coverage for introns and exons to the Drosophila_melanogaster.BDGP6.32.107.gtf annotation using the superintronic package in R with default parameters [82].

**Validation of intronic data with NET-seq data.**   In order to validate that the intronic data is representative of nascent zygotic gene transcription, NET-seq data from early embryos was used [29]. Both the NET-seq and intronic data were length normalised and a Spearman's rank correlation analysis was used to analyse the relationship.

**Modelling.**   We model dependence between pre-mRNA, $p(t)$, and mature mRNA, $m(t)$, through a GP regression that follows dynamics of an ODE of the form:

$$\frac{dm(t)}{dt} = Sp(t) - Dm(t),$$

where $p(t)$ is assumed to be a GP with RBF kernel [31,32]. This differential equation can be solved in closed form and it can be shown that $m(t)$ is also a GP with a certain kernel. For more details and specification of this kernel, we refer to S1 Supplementary methods. The results show that $m(t)$ and $p(t)$ can be modelled jointly as a GP regression with a block kernel that depends on biologically interpretable parameters such as $S$ and $D$. It is assumed that we observe $m(t)$ and $p(t)$ at discrete times with measurement noise terms that have variances $\sigma_m^2$ and $\sigma_p^2$ for mRNA and pre-mRNA, respectively. Thus, we have 6 parameters that we estimate: 2 parameters of RBF kernel ($l$ –lengthscale, $\sigma^2$–variance, which correspondingly define smoothness and amplitude of possible functions underlying pre-mRNA dynamics), 2 parameters $S$ and $D$, which describe the relationship between mRNA and pre-mRNA, and 2 measurement noise variances $\sigma_m^2$ and $\sigma_p^2$.

We assign priors to these 6 parameters and use the L-BFGS-B algorithm to find maximum a posteriori (MAP) point estimates. Further, we perform sampling from the posterior distribution using Metropolis-adjusted Langevin algorithm (MALA). By using gradient information, this algorithm allows exploring posterior distribution more efficiently than standard MCMC methods. After inferring the posterior distribution using sampling with MALA, we can quantify uncertainty about all 6 parameters of the model and uncertainty about the underlying functions for pre-mRNA and mRNA dynamics. In S1 Supplementary methods, we demonstrate the performance of the model and inference methods on simulated data: degradation rates ranging between 0.003 and 0.05 (corresponding to half-lives of 231.0 minutes and 13.8 minutes, respectively) are estimated accurately by both MAP and MALA approaches. The details of the implementation, including the assumptions on prior distributions, are discussed in S1 Supplementary methods.

From parameter estimates of $D$, half-lives were obtained using the following relationship:

$$t_{1/2} = \frac{\ln(2)}{D}.$$

Transcripts were then grouped into short, medium, and long half-life groups, setting the boundaries at 15 and 25 minutes. Any transcripts with a half-life longer than 250 minutes were removed from the dataset.

Python implementation of the model is available from https://github.com/ManchesterBioinference/GP_Transcription_Dynamics.

**Clustering.**   The intronic data ($n$ = 5,035) was z-transformed and clustered using GPclust, a package specifically designed for clustering noisy time series data using GPs [39]. Intronic clusters of interest exhibiting a range of expression profiles were selected (clusters 2 and 5) (S5 Fig). The zygotic transcripts ($n$ = 593) corresponding to the genes in each selected intronic cluster were then normalised and clustered (Figs 3 and S6). Summary statistics for the half-lives of the genes in each zygotic cluster were then computed for clusters with >2 transcripts

with estimated half-lives. A list of the transcripts for intronic clusters 2 and 5 and their corresponding zygotic clusters can be found in S4 Table.

**Analysis of time delays.** Time delays between the peaks in the pre-mRNA and mature mRNA time series from the zygotic set ($n$ = 593) were estimated by fitting a GP with RBF kernel to each time series. A total of 100 samples from each GP were taken and the delay between the peak of each sampled function of premature and mature mRNA was computed to provide estimates of the delay with uncertainty. Any transcripts with delays ≤0, or with mature mRNA profiles peaking at the final time point (t = 220), were removed. Transcripts were then grouped into short, medium, and long delay groups, setting the boundaries from the 33% and 66% quantiles of the data (17.55 and 36.16 minutes, respectively). An mRNA was classified as short if there was 90% probability that the delay for that mRNA was in the short delay interval. All statistical analysis was carried out in Python using the scipy, sklearn, and statannot libraries.

**Gene ontology analysis.** Gene ontology analysis was conducted using GOrilla [83]. Enrichment of short and long half-life mRNAs was performed using the half-life set as the target set and the entire group of dynamic genes from the RNA-seq dataset ($n$ = 8,791) as the reference set with default parameters.

**Codon usage and translation efficiency analysis.** The CSC value was calculated for each codon as previously described [43,84]. The CSC is equivalent to the Pearson correlation coefficient, calculated by plotting the frequency of each codon per transcript within our dataset against its half-life. Classification of optimal *Drosophila* codons used is in [43]. A chi-square test of association between optimal and non-optimal codons in positive and negative CSC groups was determined. The codon optimality score was determined by calculating the total proportion of optimal codons within each transcript. Transcripts were grouped by their half-life category and an independent $t$ test was used to determine significance in codon optimality between groups. Translation efficiency data was obtained from published data of 2- to 3-hour embryos [42]; 3′ UTR and transcript lengths were obtained from Flybase [85].

**Quantification of mRNA end-to-end distance.** mRNA compaction, the distance between the 5′ and 3′ ends of the transcripts, was analysed using smFISH images where the 5′ and 3′ ends are bound by probes labelled with different fluorophores. After quantifying the number and position of the mRNA ends in both channels and removing transcription sites (see Image analysis), the spot position data was analysed with a custom Python script to find optimal spot pairs by solving a paired assignment problem. The distance between $n$ 5′ spots and $m$ 3′ spots are computed and stored as a distance matrix. The optimal assignment of 5′ and 3′ pairs is then found by minimising this distance matrix to give a set of paired spots with a minimum total distance between all pairs. Spot pairs are then filtered for distances less than 300 nm where the ends are considered to be colocalised and belonging to the same RNA. This 300 nm upper threshold was selected as described in a previous study [46]. For all colocalised 5′ and 3′ spots, the distribution of distances was then analysed and summary statistics extracted.

**Analysis of mRNA colocalisation with P-bodies.** mRNA localisation within P-bodies was determined from confocal images using a custom script in Python. This script uses the position data for the mRNAs (stained with 1 set of probes) and P-bodies outputted from Airlocalize and calculates the distance between a given mRNA and every P-body. The minimum distance is then selected so that an mRNA is assigned to its closest P-body. If this distance is less than 200 nm (a typical radius of a P-body) then the RNA is considered to be colocalised with the P-body. The proportion of mRNAs located within and outside of P-bodies is then analysed to determine whether a given mRNA is enriched within P-bodies in the cytoplasm. In order to do this, we derived the P-body colocalisation index, a measure of the degree of

colocalisation with P-bodies of an mRNA of interest:

$$C_P = \frac{m_{coloc}}{m_{total} \cdot N_P}.$$

Where $C_P$ is the P-body colocalisation index, $m_{coloc}$ is the number of mRNAs colocalised with P-bodies, $m_{total}$ is the total number of mRNAs, and $N_P$ is the number of P-bodies.

For analysis of dual colour probe sets in the P-body background, the compaction analysis was done as above to identify pairs of colocalised 5′ and 3′ ends and lone ends that were either unpaired from the optimal assignment or were more than 300 nm away from the assigned pair. Colocalisation of each probe set with P-bodies was then analysed, using a more conservative threshold of 150 nm to ensure that a sufficient proportion of the mRNA was located inside the P-body, as this staining only labels the far ends of transcripts. The enrichment of unpaired 5′, 3′, and whole mRNAs in P-bodies within an image was derived by dividing the number of the species in P-bodies by the total number of the species in the image.

## Supporting information

**S1 Data. Numerical data underlying Figs 1–6 and S1–S4 and S7 and S9 and S10 and S12.**
(XLSX)

**S1 Table. RNA-seq reads for gene-level data (exons and introns) and transcripts.**
(XLSX)

**S2 Table. Table of half-lives.**
(XLSX)

**S3 Table. Table of smFISH probes used in this study.**
(XLSX)

**S4 Table. Intronic and zygotic clusters and IDs.**
(XLSX)

**S1 Supplementary methods. Filtering of the genes, model formulation, and Bayesian inference with MCMC, simulation studies.**
(PDF)

**S1 Fig. Staging of early embryos for RNA-seq.** (A) Images of embryos were captured immediately prior to collection, and the internuclear distance and embryo length were measured for each to give an internuclear ratio. (B) The internuclear ratio at nuclear cleavage cycles (nc) can be used to accurately stage embryos. (C) Timing of each nc at 20°C used in experiments. Underlying data can be found in S1 Data.
(TIF)

**S2 Fig. Read coverage and pre-mRNA profiles of early zygotic genes.** (A) Binned read coverage across exons and introns for 1 replicate for each of the time points within the time series. (B) Scatterplot of our RNA-seq reads versus NET-seq read counts showing a relationship (Spearman's Rank correlation $\rho = 0.46$, $p = 5.1 \times 10^{-11}$). Intronic read counts (RPKM) of genes that show early transcription in our dataset, grouped depending on peak expression at (C) nc12 (105 min), (D) nc13A (115 min), or (E) nc13B (125 min), based on timings at 20°C. Underlying data can be found in S1 Data.
(TIF)

**S3 Fig. Examples of alternative isoform and noncoding RNA expression during development in *Drosophila* embryos.** (A) Quantification of the 2 different transcripts (TPM) of the

*Meltrin* gene. The FBtr0301499 isoform (purple) is not detected in embryos <160 min AEL but is detected at later time points in addition to the FBtr0301498 (green) transcript. Mapped reads are shown below with a region highlighted in purple depicting the increase in the alternative transcript at time point 190 and 220 min. (B) As in (A) but for the gene *tkv*. Transcription of the FBtr0079086 (purple) and FBtr0079089 (orange) isoforms switches during the time course of development, as highlighted on the mapped reads below. Expression of noncoding RNAs, including 2 copies of tRNA-Asp and a pseudogene (CR14033), overlap the *tkv* locus as shown in the expanded region below from time point 220 min. (C) Expression of the noncoding RNAs *bithoraxoid* and *iab-8*, *iab-8* also has 2 overlapping ncRNAs. Gene-level counts (RPKM) show dynamic expression across the time course of these 2 noncoding RNAs. Gene regions for the ncRNAs (red) are shown with the genome browser tracks below. Underlying data can be found in S1 Data.
(TIF)

**S4 Fig. High embryo to embryo variation in transcript numbers masks variation due to degradation over cell division.** (A) Parameter estimates for degradation rates and credible regions of short and long half-life mRNAs shown in Fig 2D. (B) RNA-seq data for *gogo* fitted with the Gaussian process model. A half-life of 19 min is inferred for this mRNA. (C) Confocal images of pre- and post-division during the 13th and 14th nuclear cycles showing nuclei stained with DAPI (blue) and single mRNAs (white). (D) Data for the number of mRNAs per cell for embryos at various time points during the cell division (embryo timings are based on referencing live imaging movies, see Methods), fitted with an exponential function from which a half-life of 30 min was calculated. The data has a low signal to noise ratio of 0.0013, meaning that the reduction in transcript numbers over time due to degradation (signal) is much smaller than the natural embryo to embryo variation in transcript numbers (noise). (E) Theoretical data demonstrating the reduction in transcript numbers that would be expected for an mRNA with a 19 min half-life over a 210-s time frame, which is a reduction of approximately 10%. Underlying data can be found in S1 Data.
(TIFF)

**S5 Fig. Output from clustering of 5,035 intronic transcripts using GPclust.** Data for all pre-mRNAs in the cluster are shown with shaded credible regions and inferred function as a solid line. The number of pre-mRNAs in each cluster is shown in the top right corner of each plot and graphs are arbitrarily coloured. Clusters that show interesting dynamics and contain high numbers of pre-mRNAs (1, 2, 3, 5, 18, 20) are highlighted and also displayed in Fig 3Ai. Underlying data can be found in S1 and S4 Tables.
(TIFF)

**S6 Fig. Sub-clustering of zygotic transcripts with intronic data in intronic cluster 2 using GPclust.** Data for all mRNAs in intronic cluster 2 are shown with shaded credible regions and inferred function as a solid line. The number of mRNAs in each cluster is shown in the top right corner of each plot and graphs are arbitrarily coloured. The mean half-lives of the transcripts in the cluster are shown where the cluster has >2 transcripts with estimated half-lives. Underlying data can be found in S1 and S4 Tables.
(TIFF)

**S7 Fig. mRNA properties and stability.** (A) Graph showing the correlation between mRNA half-life and 3′ UTR length. Data fit with linear regression model, Pearson's r = 0.06, *p* = 0.34. (B) Correlation between mRNA length and half-life. Transcripts are coloured by half-life category short (green), medium (orange), or long (blue). Data fit with linear regression model, Pearson's r = 0.09, *p* = 0.13. (C) Transcripts were clustered based on similar codon usage using

K-means and the level of enrichment of each codon within the transcript is shown in the heatmap. The average half-life of each cluster is shown to the left. Codons are coloured by whether they are optimal (blue) vs. non-optimal (yellow) with stop codons shown in black. Underlying data can be found in S1 Data.
(TIF)

**S8 Fig. Compaction of mRNAs in early embryos.** (A) A table of the transcripts used in smFISH experiments. (B) Confocal images of fixed embryos showing smFISH detection of the 5′ (magenta) and 3′ (yellow) ends of the mRNAs used in the compaction experiments quantitated in Fig 4F. Scale bars: 5 μm.
(TIF)

**S9 Fig. Control experiments for smFISH.** (A) Proportions of 5′ (670) and 3′ (570) ends and whole mRNAs detected in smFISH experiments for all the compaction forward data described in Fig 4F. Numbers represent total mRNAs in the image (whole, 5′ only and 3′ only). Schematic illustrates the forward staining scheme. (Bi) Schematic shows full-length *otd* mRNAs detected using probes with alternating labels used as a precision control. Confocal images of fixed embryos stained with alternating smFISH probes. All images are maximum projections of 7 Z slices. Scale bars: 5 μm. (Bii) Table of detection efficiencies for the 2 mRNAs used in precision control experiments. (C) Proportions of 5′ (570) and 3′ (670) ends and whole mRNAs detected in smFISH experiments for reverse data (with switched fluorophores). Numbers represent total mRNAs in the image as in A. (D) Graph shows the end-to-end distances of mRNAs with different stabilities in the reverse fluorophore experiment to Fig 4F, data are shown in boxplots for each half-life category with $n = 3$ embryos for each transcript. (E) Proportion of *otd* signals detected in only the 570 or 670 channel, or colocalised signal in the precision or both compaction experiments. Underlying data can be found in S1 Data.
(TIFF)

**S10 Fig. Colocalisation between mRNAs of various stabilities and the P-body marker Me31B.** (A) Confocal images of early nc14 Me31B-GFP embryos showing smFISH staining of the indicated test mRNAs. DAPI marks the nuclei (blue), mRNAs are shown in magenta (magenta arrowhead), GFP-Me31B marking P-bodies in green (green arrowhead), and mRNAs colocalising with P-bodies in white (white arrowhead). All images are maximum projections of 7 Z slices of a confocal image. Scale bar: 5 μm. (B) Analysis of P-body sizes reveals an average radius of 0.2 μm. (Ci) Analysis of the correlation between Me31B binding from 2–3 hour embryos and (Cii) 3–4 hour embryos with model half-lives. A significant negative correlation ($r = -0.16$, $p = 0.014$) is found at 2–3 hours but no significant relationship is found at 3–4 hours ($r = -0.07$, $p = 0.30$). Underlying data can be found in S1 Data.
(TIFF)

**S11 Fig. Colocalisation of mRNAs and their 5′ and 3′ ends with the P-body marker Me31B.** Confocal images of early nc14 Me31B-GFP embryos showing smFISH staining of the indicated test mRNAs with 5′ and 3′ probe sets. For each test mRNA, the same region of the embryo is shown with the 5′ and 3′ mRNA probe sets separately for clarity. mRNAs are shown in magenta, GFP-Me31B marking P-bodies in green, and DAPI labelling nuclei in blue. All images are maximum projections of 7 Z slices of a confocal image. Scale bar: 2 μm.
(TIF)

**S12 Fig. Proportions of lone 5′, lone 3′, and whole mRNAs colocalised with P-bodies.** (A) Schematic demonstrating quantitation of the proportion of each species in P-bodies; 5′ and 3′ spots are detected and then paired (see Methods) to give 3 species—whole mRNAs (grey), lone

5′ ends (cyan), and lone 3′ ends (magenta). The number of each species that colocalises with P-bodies is divided by the total number of that species to give a percentage enrichment in P-bodies. (Bi) Quantification of the percentage of unpaired mRNA 5′ and 3′ ends in P-bodies relative to the total number of lone 5′ or 3′ ends in the switched probe fluorophore experiments for *otd*, *Dfd*, and *cv-2* (see also Fig 6C). Paired *t* test was used to determine significance. (Bii) As in (Bi) but alternating *otd* probes were quantified. (C) Percentages of each species (whole, lone 5′ and lone 3′) in P-bodies across the test set of mRNAs. mRNAs are ordered by their half-life from the shortest (*otd*) to the longest (*cv-2*) half-life. One-way repeated measures ANOVA was used to determine significance with α = 0.05. Underlying data can be found in S1 Data. (TIFF)

## Acknowledgments

We thank Jing Yang for processing the RNA-seq data, Lijing Lin for help with the intronic read coverage, Nuha BinTayyash for filtering the data for dynamics, Mark Ashe for helpful discussions, the University of Manchester Genomics Technology Core Facility, Fly Facility and Bioimaging Facility for support, and Mark Ashe, Sophie Frampton, and Catherine Sutcliffe for comments on the manuscript. Schematics in figures were created using BioRender.com.

## Author Contributions

**Conceptualization:** Lauren Forbes Beadle, Jennifer C. Love, Yuliya Shapovalova, Magnus Rattray, Hilary L. Ashe.

**Funding acquisition:** Magnus Rattray, Hilary L. Ashe.

**Investigation:** Lauren Forbes Beadle, Jennifer C. Love, Yuliya Shapovalova.

**Software:** Yuliya Shapovalova, Artem Artemev.

**Writing – original draft:** Lauren Forbes Beadle, Jennifer C. Love, Yuliya Shapovalova, Magnus Rattray, Hilary L. Ashe.

**Writing – review & editing:** Lauren Forbes Beadle, Jennifer C. Love, Yuliya Shapovalova, Magnus Rattray, Hilary L. Ashe.

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
