## [Editor Report · Decision Letter 0]

21 Apr 2022

Dear Dr Ashe, 

Thank you for submitting your manuscript entitled "Modelling global mRNA dynamics during Drosophila embryogenesis reveals a relationship between mRNA degradation and P-bodies" for consideration as a Research Article by PLOS Biology.

Your manuscript has now been evaluated by the PLOS Biology editorial staff as well as by an academic editor with relevant expertise and I am writing to let you know that we would like to send your submission out for external peer review.

Once your full submission is complete, your paper will undergo a series of checks in preparation for peer review. Once your manuscript has passed the checks it will be sent out for review. To provide the metadata for your submission, please Login to Editorial Manager (https://www.editorialmanager.com/pbiology) within two working days, i.e. by Apr 25 2022 11:59PM.

If your manuscript has been previously reviewed at another journal, PLOS Biology is willing to work with those reviews in order to avoid re-starting the process. Submission of the previous reviews is entirely optional and our ability to use them effectively will depend on the willingness of the previous journal to confirm the content of the reports and share the reviewer identities. Please note that we reserve the right to invite additional reviewers if we consider that additional/independent reviewers are needed, although we aim to avoid this as far as possible. In our experience, working with previous reviews does save time. 

If you would like to send previous reviewer reports to us, please email me at ialvarez-garcia@plos.org to let me know, including the name of the previous journal and the manuscript ID the study was given, as well as attaching a point-by-point response to reviewers that details how you have or plan to address the reviewers' concerns. 

Kind regards,

Ines

--

Ines Alvarez-Garcia, PhD

Senior Editor

PLOS Biology

---

## [Decision Letter · Decision Letter 1]

30 May 2022

Dear Dr Ashe,

Thank you for your patience while your manuscript entitled "Modelling global mRNA dynamics during Drosophila embryogenesis reveals a relationship between mRNA degradation and P-bodies" was peer-reviewed at PLOS Biology. It has now been evaluated by the PLOS Biology editors, an Academic Editor with relevant expertise, and by three independent reviewers. 

The reviews are attached below. As you will see, the reviewers find the conclusions interesting, however they also raise several issues that would need to be addressed with further analyses. One of the main concerns is the low number of mRNAs analysed by smFISH, and the reviewers think that it needs to be expanded to make the results robust and significant. In addition, they mention that the title and abstract are currently overstated and should be toned down, and that the modelling needs to improve. In light of the reviews and after discussing them with the Academic Editor, we have decided to invite you to revise the work to thoroughly address the reviewers' comments.

Given the extent of revision needed, we cannot make a decision about publication until we have seen the revised manuscript and your response to the reviewers' comments. Your revised manuscript is likely to be sent for further evaluation by all or a subset of the reviewers.

**IMPORTANT - SUBMITTING YOUR REVISION**

3. Resubmission Checklist

a) *PLOS Data Policy*

b) *Published Peer Review*

d) *Blurb*

Please also provide a blurb which (if accepted) will be included in our weekly and monthly Electronic Table of Contents, sent out to readers of PLOS Biology, and may be used to promote your article in social media. The blurb should be about 30-40 words long and is subject to editorial changes. It should, without exaggeration, entice people to read your manuscript. It should not be redundant with the title and should not contain acronyms or abbreviations. For examples, view our author guidelines: https://journals.plos.org/plosbiology/s/revising-your-manuscript#loc-blurb

Sincerely,

Ines

--

Ines Alvarez-Garcia, PhD

Senior Editor

PLOS Biology

Reviewers' comments

Rev. 1:

In this manuscript, the authors derive mRNA half-life data from select mRNAs in early Drosophila embryogenesis taking advantage of the large changes in gene expression that occur when embryos switch to zygotic transcription. They analyze their data and correlate them with existing ribosome profiling results and other mRNA features. They conclude that mRNA half-life does not depend on translation efficiency but short-lived mRNAs tend to be more strongly associated with cytoplasmic P-bodies, membraneless organelles previously implicated in mRNA storage but also in mRNA decay. Overall, this is an interesting manuscript that should be of relevance to a broad readership interested in the post-transcriptional regulation of gene expression. However, there are also some significant weaknesses that need to be addressed prior to publication.

1. The main experimental weakness is the small number of mRNAs that are examined by smFISH in Figures 4 and 5. I am aware that these are likely technically involved experiments but these data sets need to be expanded in order to make them robust and meaningful. Currently, there is a large bias towards long-lived mRNAs and the authors need to add 2-3 additional short and medium half-life mRNAs each to confirm the current trends.

2. Is there a bias towards 3' end sequences in short-lived mRNAs in their global sequencing reads as their P-body association data would suggest?

3. The writing of the manuscript needs to be improved:

a. The abstract overstates the conclusions and it should be clearly mentioned that the half-lives are derived from changes in intronic reads. It is also not correct that they can determine half-lives 'globally'. In addition, it must be clear from the abstract that their conclusions are based on correlations (or lack thereof).

b. There is no need to (overly) negatively discuss other approaches to measure half-lives such as metabolic labeling. Yes, the labeling times etc can introduce errors in metabolic approaches but it has been a powerful method to determine decay rates on a global scale in various systems. Technical limitations in the embryo fully together with the non-invasive nature justify their approach (but it also has limits and will not be generally applicable, which should be discussed).

c. P13. The authors wanted to test "whether mRNA localisation influences stability". Again, the experiments allow them to correlate only and maybe stability influences localization instead…

Rev. 2:

In this manuscript, the authors study the degradation of mRNAs during Drosophila embryogenesis. By staging embryos beginning at nuclear cycle 11 before zygotic transcription is turned on, single embryos were collected at intervals through nc11-14 and ending at cephalic furrow formation. RNA was isolated from these time points and intron and exon reads were used to measure changes in RNA levels during development. Since zygotic transcription is activated during their time course, they are able to identify transcriptions with intronic signals to specifically isolate transcripts whose dynamics can be followed from those mRNAs that are also maternally deposited. This data was then modelled using a Gaussian process to estimate transcript half-lives from the time-dependent changes in intronic and exonic reads. This allowed them to measure the stabilities of 279 transcripts (187 genes) and the differing intronic and exonic expression patterns were clustered that revealed that transcripts with similar intronic dynamics displayed varying exonic dynamics suggesting that this resulted from differing mRNA half-lives. To determine if translation might influence mRNA stability, smFISH targeted to the 5' and 3' ends of transcripts was used to measure the compaction of mRNAs, which has previously been shown to correlate with active translation (elongated mRNAs are translating, while compact mRNAs are inactive). The authors further characterized the position of the 5' and 3' ends with respect to P-bodies and determined that unpaired 3' ends were enriched in P-bodies.

In general, I think the findings of this paper are interesting and could be an important contribution to the field, but there are some technical questions that should be addressed prior to publication

1. It was not clear to me how pre-mRNA transcript numbers were determined from their sequencing data. How are they accounting for the differences in length of introns and exons since a single intron could give rise to many reads? Are the intronic reads evenly distributed? Are they assuming all introns are co-transcriptionally spliced and then rapidly degraded? I find it confusing that pre-mRNA levels are always higher than mature mRNA levels in Figure 2Di and Dii and believe this requires some clarification

2. In the modeling, they use a parameter S for splicing efficiency but I am not sure how this is defined. Do they assume every intron has the same splicing efficiency or all the introns of a particular transcript have the same efficiency? Can this be justified?

3. The authors claim that they are unable to perform intron and exon smFISH to determine mRNA half-lives because the embryo is not at steady state, but it is not clear to me how the measurements would be different from their sequencing data. Shouldn't it be possible to model it in a similar way?

4. In figure 4D, they show smFISH for the 5' and 3' ends of Dfd. They should also report what the total number of 5' spots, 3' spots, co-localization and detection efficiency. In the representative images, the detection efficiency looks to be low in Figure 4Dii. It also surprises me that the FISH signal is so different in size and intensity in Figure 4Dii? Is this an artifact of deconvolution? The FISH signal looks more uniform in Sup Fig S8D and E though.

5. In figure 5, the authors show the co-localization of different mRNAs with P-bodies. As Neu2 (unstable) and Dfd (stable) hare similar co-localization indexes, I find it somewhat overstated to claim that unstable mRNAs are enriched in P-bodies given the small number of mRNAs tested and the similarity stated above. There should at least be equal numbers of transcripts in each class for comparison.

6. In Figure 6, the authors look at the co-localization of 5' and 3' ends of mRNAs with P-bodies to determine if decay is occurring in P-bodies. They need to also report the number of unpaired 5' and 3' ends in the cytosol as well and the detection efficiency in order to further support their observations. For example, if they are systematically under-detecting the 5'-ends, this would result in similar data, but with a very different conclusion.

Minor comments.

1. I would suggest reformulating the title since modeling of mRNA stabilities from the RNA sequencing data did not reveal any about the location of mRNA decay and this was found by microscopy. I understand the desire to incorporate both aspects of the experiments performed in the title, but I find it to be confusing

2. The timings of collecting the embryos is not clear to me. In the text, it states that embryos were collected at 5 min intervals during nc11-13 and 15 min intervals during nc13 and nc14, however, this time is very different from what is shown in Fig1A and Table 1. This might be obvious to a Drosophila specialist because an additional time delay is implied, but requires a little more explanation for non-experts

Rev. 3:

Review: Beadle, Love et al., "Modeling global mRNA dynamics during Drosophila embryogenesis reveals a relationship between mRNA degradation and P-bodies"

The authors present an exciting method to retrieve mRNA half-lives during Drosophila zygotic genome activation. For this they use a high-resolution RNA-seq time series, performed on carefully staged single Drosophila embryos. To estimate mRNA half-lives from these datasets, they employ a Gaussian process regression model, which jointly models pre-mRNA and mature mRNA. Having estimated mRNA half-life for ~300 zygotic transcripts, the authors then investigate the contribution of various steps of mRNA lifecycle to their stability. First, they use clustering and modeling to reveal that the time delay between peak pre-mRNA and mature mRNA levels could be used as a proxy to estimate mRNA stability. Next, they show that mRNA stability is not correlated with 3'UTR length, translation efficiency or codon optimality, but is moderately related to mRNA compaction. Finally, the authors assess the relationship between mRNA half-lives, their degradation and localization in p-bodies. For this, they focus on 7 zygotic transcripts, and employ smFISH labeling and quantitative imaging. For this small set of mRNAs, they show that unstable mRNAs (short half-lives) tend to locate in ME31B positive p-bodies, where they seem to be degraded.

The major strength of the manuscript lies in:

1) the exhaustive database of mRNA half-lives (very useful to the community), along with the methodology that could be easily transferred to other systems.

2) the description of correlations between mRNA half-lives and various post-transcriptional processes. I am less convinced by the part on p-bodies which primarily relies on a small set of examined mRNAs (see discussion below).

Apart from this last part, conclusions of the manuscript are well supported by the experimental data and bioinformatic analyses.

Conclusions of this paper are of immediate interest to many people in the field of gene expression and mRNA biogenesis in vivo. I recommend this paper to be published with minor corrections.

Here are my specific questions and recommendations (very minor):

#1.ME31B

I find the title too strong, as the major finding of the manuscript is not the link between mRNA degradation and p-bodies. Actually, a correlation between localization in p-bodies and mRNA half-life is shown for a handful of transcripts (=7). In the absence if experiments manipulating ME31B, this result remains correlative and should be interpreted as such.

Similarly, I would nuance the conclusions in the results part, line 334. This is valid for 7 transcripts.

To strengthen the observation that short-lived mRNAs tend to be localized within ME31B positive p-bodies, the authors could analyze ME31B RIP-seq datasets (if available for early embryos).

#2. Modeling

The description of the Gaussian model is too vague in the main text. It is important to clearly state its assumptions and potential caveats in biological terms. Would this methodology tend to over-estimate or under-estimate mRNA half-lives?

Which temporal scale would be unreachable by this method?

Why choose the threshold 0.4 (line 139)? What would we gain/lose by being more/less stringent?

Simulations might be useful to clarify these points.

In terms of modeling, what are the advantages of using a Gaussian compared to other models?

This is briefly mentioned in the discussion (line 380) but could be elaborated on a bit more.

#3. mRNA half-life dataset

Given that intronless zygotic transcripts could not be studied with this method, could the author state the % of total zygotic transcripts Table S1 represents (line 147)?

How does intron size affect this pipeline and the output?

Part of the method assumes that pre-mRNA profiles reflect transcription dynamics. Could the authors use the recently published NET-seq results in early staged embryos to back up this assumption?

Minor points

-The quality of the images Figure 1, S1A and could be improved, as well as their sizes.

-The number of 'pairs' should be indicated in the figure legends of smFISH data using 3' and 5' probes.

-Figure 3: I would define the notion of 'delay' within Figure 3.

---

## [Decision Letter · Decision Letter 2]

7 Nov 2022

Dear Dr Ashe,

Thank you for your patience while we considered your revised manuscript entitled "Single molecule imaging and modelling of mRNA decay dynamics in the Drosophila embryo" for publication as a Research Article at PLOS Biology. This revised version of your manuscript has been evaluated by the PLOS Biology editors, the Academic Editor and the three original reviewers.

Based on the reviews (attached below), we are likely to accept this manuscript for publication, provided you address the data and other policy-related requests stated below.

In addition, we would like you to consider a suggestion to improve the title, perhaps by highlighting one of the findings. For example:

"Single molecule analysis of mRNA decay dynamics in the Drosophila embryo uncovers P-bodies 5’ to 3’ degradation"

We expect to receive your revised manuscript within two weeks. 

*Published Peer Review History*

*Press*

Sincerely,

Ines

--

Ines Alvarez-Garcia, PhD

Senior Editor

PLOS Biology

Fig. 1B-E; Fig. 2B-D; Fig. 3A-D; Fig. 4A-C, Fig. 5C, D; Fig. 6C; Fig. S1B, C; Fig. S2A-E; Fig. S3A-C; Fig. S4B, D, E; Fig. S5; Fig. S6; Fig. S7A-C; Fig. S9A, C-E; Fig. S10B, C and Fig. S12B, C

**Please also ensure that the RNA-seq data deposited in ArrayExpress (E-MTAB-11580) is publicly available at this stage.

Reviewers' comments

Rev. 1: 

The authors have addressed my (and also the other reviewers') major concerns. This has improved this interesting manuscript, and I can now recommend publication.

Rev. 2:

The authors have answered my questions and I believe it should now be published.

Rev. 3: Mounia Lagha – note that this reviewer has signed her review

The authors have answered to all of my concerns. The manuscript is now suitable for publications, Congrats

---

## [Editor Report · Decision Letter 3]

13 Dec 2022

Dear Dr Ashe,

Thank you for the submission of your revised Research Article entitled "Combined modelling of mRNA decay dynamics and single molecule imaging in the Drosophila embryo uncovers a role for P-bodies in 5’ to 3’ degradation" for publication in PLOS Biology. On behalf of my colleagues and the Academic Editor, Simon Bullock, I am delighted to say that we can in principle accept your manuscript for publication, provided you address any remaining formatting and reporting issues. These will be detailed in an email you should receive within 2-3 business days from our colleagues in the journal operations team; no action is required from you until then. Please note that we will not be able to formally accept your manuscript and schedule it for publication until you have completed any requested changes.

PRESS

Sincerely, 

Ines

--

Ines Alvarez-Garcia, PhD

Senior Editor

PLOS Biology
